# The C-terminal helix 9 motif in rat cannabinoid receptor type 1 regulates axonal trafficking and surface expression

Alexandra Fletcher-Jones, Keri L Hildick, Ashley J Evans, Yasuko Nakamura, Kevin A Wilkinson, Jeremy M Henley*

School of Biochemistry, Centre for Synaptic Plasticity, University of Bristol, Bristol, United Kingdom

**Abstract** Cannabinoid type one receptor (CB1R) is only stably surface expressed in axons, where it downregulates neurotransmitter release. How this tightly regulated axonal surface polarity is established and maintained is unclear. To address this question, we used time-resolved imaging to determine the trafficking of CB1R from biosynthesis to mature polarised localisation in cultured rat hippocampal neurons. We show that the secretory pathway delivery of CB1R is axonally biased and that surface expressed CB1R is more stable in axons than in dendrites. This dual mechanism is mediated by the CB1R C-terminus and involves the Helix 9 (*H9*) domain. Removal of the *H9* domain increases secretory pathway delivery to dendrites and decreases surface stability. Furthermore, CB1R$^{\Delta H9}$ is more sensitive to agonist-induced internalisation and less efficient at downstream signalling than CB1R$^{WT}$. Together, these results shed new light on how polarity of CB1R is mediated and indicate that the C-terminal *H9* domain plays key roles in this process.
DOI: https://doi.org/10.7554/eLife.44252.001

*For correspondence:
j.m.henley@bristol.ac.uk

Competing interests: The authors declare that no competing interests exist.

## Introduction

CB1R is one of the most abundant G-protein-coupled receptors (GPCRs) in the CNS and endocannabinoid signalling through CB1R is a neuromodulatory system that influences a wide range of brain functions including pain, appetite, mood, and memory (*Soltesz et al., 2015*; *Lu and Mackie, 2016*). Furthermore, CB1R function and dysfunction are implicated in multiple neurodegenerative disorders (*Basavarajappa et al., 2017*). Thus, modulation of endocannabinoid pathways is of intense interest as a potential therapeutic target (*Reddy, 2017*).

CB1R is present in both excitatory and inhibitory neurons, and also in astroglia, where it plays important roles in synaptic plasticity and memory (*Han et al., 2012*; *Robin et al., 2018*; *Busquets-Garcia et al., 2018*). In hippocampal neurons, ~80% of CB1R is present in intracellular vesicular clusters in the soma and dendrites (*Leterrier et al., 2006*). Strikingly, however, CB1R is not stably surface expressed on somatodendritic plasma membrane. Rather, it has a highly polarised axonal surface expression (*Irving et al., 2000*; *Coutts et al., 2001*) where it acts to attenuate neurotransmitter release (*Katona, 2009*) and modulate synaptic plasticity (*Lu and Mackie, 2016*).

How this near exclusive axonal surface expression of CB1R is established remains the subject of debate. One suggestion is that high rates of endocytosis due to constitutive activity selectively remove CB1Rs from the somatodendritic plasma membrane, resulting in an accumulation at the axonal surface (*Leterrier et al., 2006*). These internalised somatodendritic CB1Rs may then be either sorted for degradation or recycled to axons via a transcytotic sorting pathway (*Simon et al., 2013*). Alternatively, newly synthesized CB1Rs may be constitutively targeted to lysosomes, but under appropriate circumstances the CB1Rs destined for degradation are retrieved and rerouted to axons (*Rozenfeld and Devi, 2008*; *Rozenfeld, 2011*).

**eLife digest** The brain contains around 100 billion neurons that are in constant communication with one another. Each consists of a cell body, plus two components specialized for exchanging information. These are the axon, which delivers information, and the dendrites, which receive it. This exchange takes place at contact points between neurons called synapses. To send a message, a neuron releases chemicals called neurotransmitters from its axon terminals into the synapse. The neurotransmitters cross the synapse and bind to receptor proteins on the dendrites of another neuron. In doing so, they pass on the message.

Cannabinoid type 1 receptors (CB1Rs) help control the flow of information at synapses. They do this by binding neurotransmitters called endocannabinoids, which are unusual among neurotransmitters. Rather than sending messages from axons to dendrites, endocannabinoids send them in the opposite direction. Thus, it is dendrites that release endocannabinoids, which then bind to CB1Rs in axon terminals. This backwards, or 'retrograde', signalling dampens the release of other neurotransmitters. This slows down brain activity, and gives rise to the 'mellow' sensation that recreational cannabis users often describe.

Like most other proteins, CB1Rs are built inside the cell body. So, how do these receptors end up in the axon terminals where they are needed? Are they initially sent to both axons and dendrites, with the CB1Rs that travel to dendrites being rerouted back to axons? Or do the receptors travel directly to the axon itself? Fletcher-Jones et al. tracked newly made CB1Rs in rat neurons growing in a dish. The results revealed that the receptors go directly to the axon, before moving on to the axon terminals. A specific region of the CB1R protein is crucial for sending the receptors to the axon, and for ensuring that they do not get diverted to the dendrite surface. This region stabilizes CB1Rs at the axon surface, and helps to make the receptors available to bind endocannabinoids.

CB1Rs also respond to medical marijuana, a topic that continues to generate interest as well as controversy. Activating CB1Rs could help treat a wide range of diseases, such as chronic pain, epilepsy and multiple sclerosis. Future studies should build on our understanding of CB1Rs to explore and optimize new therapeutic approaches.

DOI: https://doi.org/10.7554/eLife.44252.002

Surprisingly, a direct role for the 73-residue intracellular C-terminal domain of CB1R (ctCB1R) in axonal/somatodendritic trafficking or polarised surface expression has not been identified. It has, however, been reported that motifs within ctCB1R are required for receptor desensitization and internalization (*Hsieh et al., 1999*; *Jin et al., 1999*) (reviewed by *Mackie, 2008*). Interestingly, there are two putative amphipathic helical domains in ctCB1R (*H8* and *H9* [*Ahn et al., 2009*]). *H8* has been proposed to play a role in ER assembly and/or exit during biosynthesis (*Ahn et al., 2010*; *Stadel et al., 2011*). The role of the 21-residue *H9* motif is unknown, although analogous regions have been reported to act as a $G_{\alpha q}$-binding site in both squid rhodopsin (*Murakami and Kouyama, 2008*) and bradykinin receptors (*Piserchio et al., 2005*).

Here we systematically investigated how axonal surface polarity of CB1R arises by tracking newly-synthesised CB1Rs through the secretory pathway to their surface destination. We demonstrate that a population of CB1R is preferentially targeted to the axon through the biosynthetic pathway. CB1Rs that reach the dendritic membrane are rapidly removed by endocytosis whereas CB1Rs surface expressed on the axonal membrane have a longer residence time. We further show that the putative helical domain *H9* in ctCB1R plays a key role in CB1R surface expression and endocytosis in hippocampal neurons. Taken together our data suggest that CB1R polarity is determined, at least in part, by a novel determinant in the C-terminus of CB1R that contributes to targeted delivery to the axonal compartment and the rapid removal of CB1Rs that reach the somatodendritic membrane.

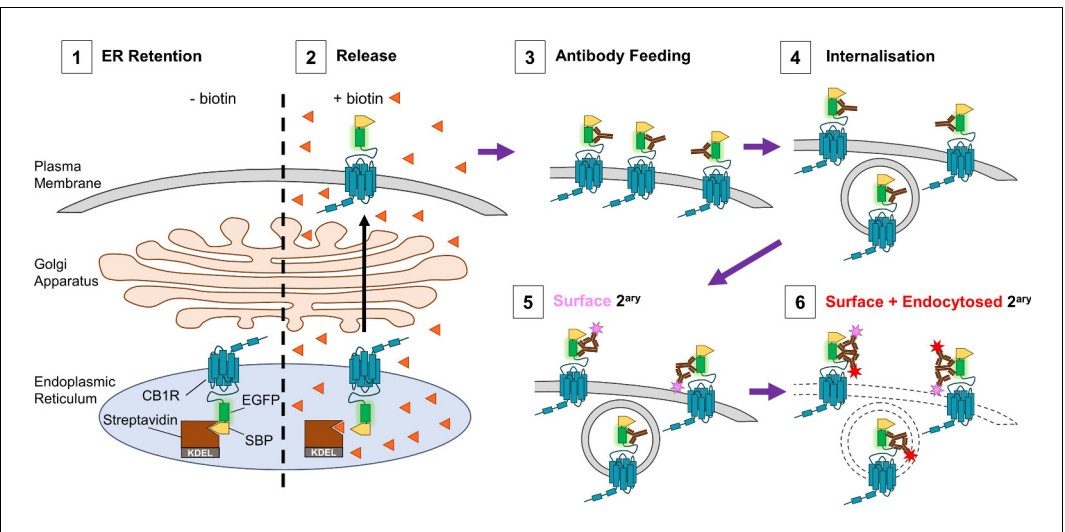

**Figure 1.** Schematic of RUSH assay and antibody feeding protocol. (1) Before the addition of biotin, SBP-EGFP-CB1R is retained in the ER by a streptavidin-KDEL hook (0 min). (2) Addition of biotin (orange triangles) releases the receptor and it begins to accumulate at the surface. (3) Antibody feeding with anti-GFP antibodies during biotin-mediated release labels newly delivered, surface expressed SBP-EGFP-CB1R. (4) A proportion of receptors internalise, still bound to primary antibody. (5) Cells are cooled to 4°C to prevent further internalisation. Live secondary antibody incubation labels retained surface receptors (indicated by magenta star). (6) After fixation and permeabilization, incubation with a different secondary antibody labels all receptors delivered to the surface during the time course of the experiment (red star = surface + endocytosed).

DOI: https://doi.org/10.7554/eLife.44252.003

## Results

### Preferential delivery of newly synthesized CB1Rs to, and retention at, the axonal membrane establishes surface polarisation

To investigate how CB1R surface polarity is established we used the retention using selective hooks (RUSH) system (*Boncompain et al., 2012*) and antibody feeding techniques to examine its secretory pathway trafficking and surface expression (*Figure 1*). We used CB1R tagged at the N-terminus with streptavidin binding peptide (SBP) and EGFP (SBP-EGFP-CB1R). When co-expressed with a Streptavidin-KDEL 'hook' that localises to the lumen of the Endoplasmic Reticulum (ER), SBP-EGFP-CB1R is anchored at the ER membrane. The retained SBP-EGFP-CB1R can then be synchronously released by addition of biotin and its trafficking through the secretory pathway and surface expression in both axons and dendrites can be monitored (*Evans et al., 2017*).

### CB1R is directly trafficked to the axon through the secretory pathway

We first examined the synchronous trafficking of total SBP-EGFP-CB1R in the somatodendritic and axonal compartments of primary hippocampal neurons (*Figure 2A–C*). Prior to biotin-mediated release, SBP-EGFP-CB1R was retained in the ER in the soma and dendrites but was absent from the axonal compartment and was not present at the cell surface (0 min; *Figure 2A*). After addition of biotin, SBP-EGFP-CB1R moved through the secretory pathway and entered the proximal segment of the axonal compartment at 25 min and continued to accumulate until 45 min when it reached its peak, which was comparable to an unretained control (O/N) (*Figure 2B–C*). These data suggest that once released from the ER, CB1R is immediately trafficked towards the axonal compartment, and passes through the axon initial segment (AIS), which constitutes an exclusion and diffusion barrier to separate the axonal from the somatodendritic compartments, via the intracellular secretory pathway.

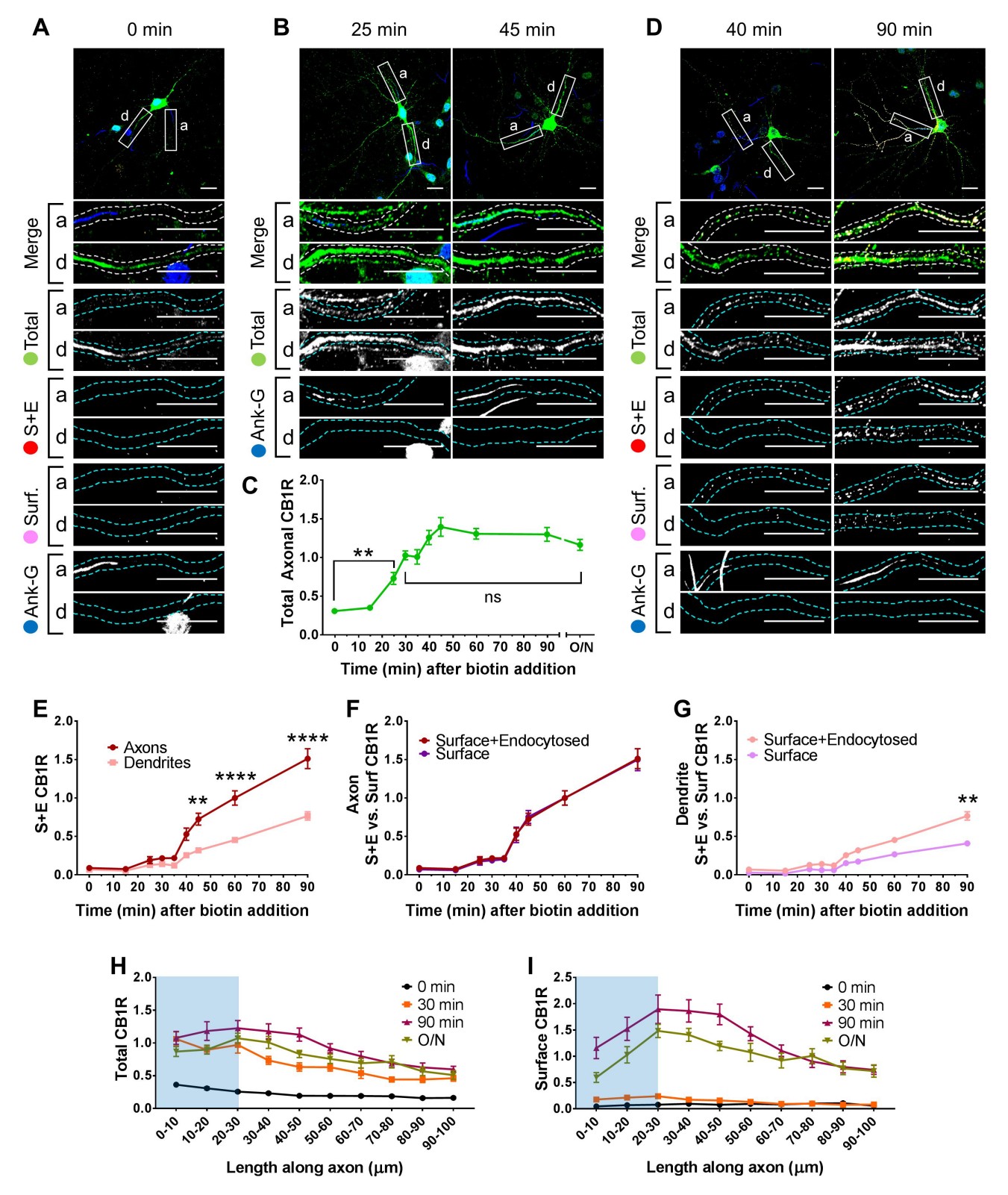

**Figure 2.** Newly synthesized CB1Rs are preferentially delivered to, and retained at, the axonal membrane to establish surface polarisation. The trafficking of SBP-EGFP-CB1R following release with biotin was monitored after 0 (no biotin), 15, 25, 30, 35, 40, 45, 60, 90 min, and overnight (O/N; non-retained control) in DIV 13 hippocampal neurons. Upper panels for each condition show whole cell field of view and lower panels are enlargements of axonal (a) and dendritic (d) ROIs. Green = total; red = surface + endocytosed; magenta = surface; blue = axon marker (Ankyrin-G). In all images the

*Figure 2 continued on next page*

*Figure 2 continued*

scale bar = 20 μm. (A) Representative image of a hippocampal neuron expressing the RUSH construct SBP-EGFP-CB1R without biotin (0 min). SBP-EGFP-CB1R is anchored in the ER of the somatodendritic compartment and is not detected in the proximal 50 μm of axons or on the surface of dendrites. Merge: green = total; blue = Ankyrin G; red = surface + endocytosed; magenta = surface. (B) Representative confocal images of total SBP-EGFP-CB1R expressed in DIV 13 hippocampal neurons 25 min and 45 min after biotin release from the ER showing that SBP-EGFP-CB1R has entered the proximal axonal compartment (initial 50 μm). Merge: green = total; blue = Ankyrin G. (C) Quantification of data represented in (A and B). SBP-EGFP-CB1R was initially absent from the axon but entered after 25 min and continued to accumulate until it plateaued after 45 min to a level comparable to a non-retained control (O/N). One-way ANOVA with Tukey's *post hoc* test. N = three to six independent experiments, n = 19–45 neurons per condition. 0 min vs. 25 min: mean ± SEM, 0.307 ± 0.0173 vs. 0.729 ± 0.0772; N = 6, n = 45 vs. N = 3, n = 19; **p = 0.0018. 30 min vs. ON: mean ± SEM, 1.03 ± 0.0597 vs. 1.2 ± 0.0632; N = 4, n = 32 vs. N = 4, n = 24, [ns]p = 0.8186. (D) Representative confocal images of total and surface expressed SBP-EGFP-CB1R in DIV 13 hippocampal neurons 40 min and 90 min after biotin-mediated release showing that SBP-EGFP-CB1R is preferentially delivered to, and retained at, the axonal surface. Merge: surface to total seen as white; endocytosed to total seen as yellow. (E) Quantification of data represented in (D). SBP-EGFP-CB1R reached the proximal surface of the axon 40 min after release and the surface of dendrites 60 min after release. Furthermore, significantly more SBP-EGFP-CB1R reached the axonal versus dendritic surface at 45, 60, and 90 min. 45 min, Axons vs. Dendrites: mean ± SEM, 0.723 ± 0.077 vs. 0.319 ± 0.035; N = 3, n = 20 vs. N = 3, n = 20; **p = 0.0054. 60 min, Axons vs. Dendrites: mean ± SEM, 1.00 ± 0.093 vs. 0.452 ± 0.023; N = 6, n = 46 vs. N = 6, n = 46; ****p < 0.0001. 90 min, Axons vs. Dendrites: mean ± SEM, 1.511 ± 0.129 vs. 0.566 ± 0.054; N = 4, n = 26 vs. N = 4, n = 26; ****p < 0.0001. (F) Quantification of data represented in (D). Comparison between surface + endocytosed (red; see E) and surface (magenta) curves show that SBP-EGFP-CB1R was retained on the surface of axons. (For all p > 0.9999). (G) Quantification of data represented in (D). Comparison between surface + endocytosed (pale red; see (E) and surface (pale magenta) curves show that SBP-EGFP-CB1R was internalised from the surface of dendrites. 90 min, SE vs. S: mean ± SEM, 0.766 ± 0.054 vs. 0.408 ± 0.038; N = 4, n = 26 vs. N = 4, n = 26; **p = 0.0046. Statistical analyses in (E-G); Two-way ANOVA with Tukey's *post hoc* test (all analysed and corrected for multiple comparisons together). Three to six independent experiments, n = 19–45 neurons per condition. (H) Distribution of total CB1R along the first 100 μm of the axon indicates that CB1R is trafficked within the axon. By 30 min after release from the ER, and before CB1R reaches the surface, CB1R is present at least 100 μm away from the soma at levels comparable to an unretained control (O/N). The blue shaded portion indicates the location of the AIS (defined by Ankyrin-G immunostaining). Two-way ANOVA with Sidak's *post hoc* test. Four to six independent experiments, n = 12–18. 90–100 μm, 0 vs. 30 min: mean ± SEM, 0.163 ± 0.014 vs. 0.452 ± 0.045; N = 6, n = 12 vs. N = 4, n = 18; *p = 0.0243. 90–100 μm, 30 min vs. O/N: mean ± SEM, 0.452 ± 0.045 vs. 0.511 ± 0.066; N = 4, n = 18 vs. N = 4, n = 15; [ns]p = 0.905. (I) Distribution of surface expressed CB1R along the first 100 μm of the axon shows an accumulation of CB1R at the distal region of the AIS 90 min after release from the ER. This accumulation in the AIS is reduced in the O/N unretained control consistent with lateral diffusion within the membrane. Two-way ANOVA with Sidak's *post hoc* test. Four to six independent experiments, n = 12–18. 0–50 μm, 90 min vs. O/N: All points p ≤ 0.0285. 50–100 μm, 90 min vs. O/N: All points p ≥ 0.0878.

DOI: https://doi.org/10.7554/eLife.44252.004

### *De novo* CB1R is more rapidly surface expressed in axons than in dendrites

Having established that SBP-EGFP-CB1R released from the ER traffics directly to axons, we next investigated where and when the newly synthesised SBP-EGFP-CB1R first reaches the plasma membrane. We determined how much SBP-EGFP-CB1R was surface expressed during a given time period using an antibody feeding assay (*Evans et al., 2017*) (*Figure 1*). Antibody feeding was performed concurrent with the addition of biotin to release ER-retained SBP-EGFP-CB1R. This protocol labels both surface expressed CB1Rs and those that have been surface expressed and subsequently endocytosed (*Figure 1*; *Figure 2D–G*; surface + endocytosed), giving a measure of total amount of surface expression irrespective of internalisation. SBP-EGFP-CB1R was surface expressed in the proximal segment of axons 40 min after release from the ER, whereas in dendrites, CB1R was not surface expressed until 60 min after release (*Figure 2E*). Moreover, significantly more SBP-EGFP-CB1R reached the surface of axons than the surface of dendrites 45, 60, and 90 min after release from the ER (*Figure 2E*). These data demonstrate that the secretory pathway delivers a greater amount of CB1R more rapidly to the axonal membrane than to the dendritic membrane.

### *De novo* CB1R is retained longer at the surface of axons than of dendrites

It has been suggested CB1R polarity is maintained by differential rates of endocytosis in the somatodendritic and axonal compartments (*Leterrier et al., 2006*; *McDonald et al., 2007a*). To test this, we also stained for surface SBP-EGFP-CB1R and compared the amount of surface expressed SBP-EGFP-CB1R to the amount of surface + endocytosed SBP-EGFP-CB1R in axons (*Figure 2D,F*) and dendrites (*Figure 2D,G*). In the proximal segment of axons, the normalised surface and surface + endocytosed curves were identical, suggesting that most surface expressed SBP-EGFP-CB1R is stable and retained at the axonal membrane (*Figure 2D,F*). This may be due either to minimal endocytosis or to the efficient recycling of endocytosed receptors. In stark contrast, however, in

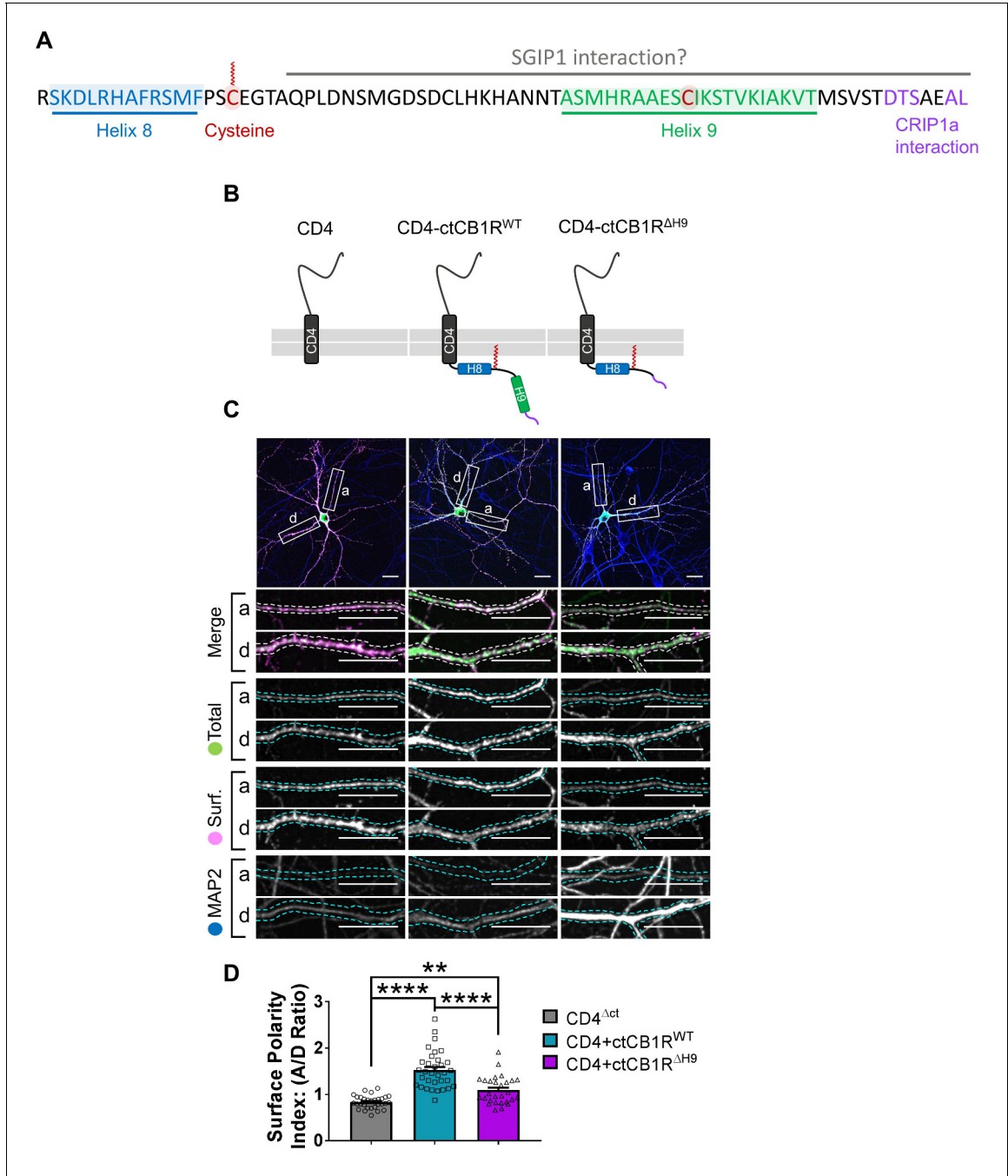

**Figure 3.** The C-terminal domain of CB1R, especially the Helix 9 motif, plays a role in axonal surface polarisation. (**A**) Amino acid sequence of the C-terminus of rat CB1R highlighting helical motifs (Helix 8 and Helix 9) predicted from NMR spectroscopy and computational modelling. The proposed interaction domains of CRIP1a and SGIP1, and the palmitoylated cysteine residue involved in membrane association and G-protein coupling, are also indicated. A potentially post-translationally modified cysteine in *H9* is highlighted. (**B**) Schematic of the CD4-ctCB1R chimeric proteins used. (**C**) Representative confocal images of hippocampal neurons showing the distribution of expressed CD4 (left), CD4-ctCB1R$^{WT}$ (middle), or CD4-ctCB1R$^{\Delta H9}$ (right). Upper panels for each condition show a whole cell field of view and lower panels are enlargements of axonal (**a**) and dendritic (**d**) ROIs. Green = total; magenta = surface; blue = dendrite marker (MAP2). Merge: surface to total seen as white. Scale bar = 20 μm. (**D**) Quantification of data represented in (**C**) presented as the surface polarity index (A/D ratio). CD4-ctCB1R$^{WT}$ strongly favoured the axonal compartment compared to CD4 alone. CD4-ctCB1R$^{\Delta H9}$ favoured the axonal compartment significantly less than CD4-ctCB1R$^{WT}$. One-way ANOVA with Tukey's *post hoc* test. N = three independent experiments; n = 28–33 neurons per condition. CD4 vs. WT: mean ± SEM, 0.834 ± 0.0255 vs. 1.52 ± 0.0696; N = 3, n = 30 vs. N = 3, n = 33; ****p < 0.0001. CD4 vs. ΔH9: mean ± SEM, 0.834 ± 0.0255 vs. 1.09 ± 0.0562; N = 3, n = 30 vs. N = 3, n = 28; **p = 0.0050. WT vs. ΔH9: mean ± SEM, 1.52 ± 0.0696 vs. 1.09 ± 0.0562; N = 3, n = 33 vs. N = 3, n = 28; ****p < 0.0001.

DOI: https://doi.org/10.7554/eLife.44252.005

dendrites there is significantly less surface than surface + endocytosed SBP-EGFP-CB1R 90 min after addition of biotin, indicating that surface expressed CB1R is more rapidly endocytosed from and/or not recycled back to the dendritic membrane (*Figure 2G*).

## CB1R is trafficked to more distal parts of the axon via intracellular mechanisms

Analysis of CB1R total fluorescence along the axon indicates that by 30 min after release from the ER, and before CB1R appears on the surface, intracellular CB1R has progressed through the AIS and is already present at least 100 µm along the axon at levels similar to the unretained control (O/N; *Figure 2H,I*). These data indicate that CB1R-containing secretory vesicles can rapidly travel to more distal areas of the axon. Furthermore, surface CB1Rs delivered from the secretory pathway accumulate at the final portion of the AIS, before then progressing further along the axon (*Figure 2I*). Interestingly, 100 µm along the axon CB1R levels reach a steady state 90 min after release, at levels similar to when receptors are released overnight. However, following overnight release fewer receptors remain in the most proximal region of the axon. These results suggest that additional mechanisms contribute to the delivery of CB1R receptors the distal axon and presynaptic boutons. This process occurs over a time-course of several hours and could involve lateral surface diffusion and trapping analogous to the accumulation of AMPARs at the postsynaptic membrane (*Borgdorff and Choquet, 2002*).

## A two-part model of CB1R polarity

Our results using RUSH time-resolved analysis show that CB1R surface polarity is initially established and maintained by two distinct but complementary mechanisms. Firstly, we show the novel finding that the secretory pathway preferentially delivers CB1R to the axonal surface, with significantly less going to the dendritic surface. Secondly, by distinguishing between surface and surface + endocytosed receptors, our antibody feeding experiments show that newly delivered CB1R is preferentially retained/stabilised at the axonal membrane and internalised from the dendritic membrane. Previous literature proposes that this differential internalisation is due to the presence of agonist in the dendritic membrane and absence of agonist on axonal membrane (*Leterrier et al., 2006*; *Ladarre et al., 2014*), although a potential role for constitutive internalisation distinct to agonist-induced internalisation has also been proposed (*McDonald et al., 2007a*). Taken together, we propose that preferential delivery to the proximal segment of the axon and less rapid internalisation of axonally surface expressed CB1Rs are major contributors to the axonal surface polarisation of CB1R in hippocampal neurons.

## *H9* contributes to axonal surface polarisation

The intracellular ctCB1R is implicated in desensitization and internalization (reviewed in *Mackie, 2008*; *Stadel et al., 2011*) and structural motifs and potential interaction partners have been identified (*Figure 3A*; *Stadel et al., 2011*). However, the role of this region in determining axonal polarity has not been investigated and the function of the *H9* structural motif is unknown. We therefore wondered whether ctCB1R, and *H9* in particular, contribute to CB1R surface polarisation.

To test the role of the C-terminal domain we initially used CD4, a single-pass membrane protein that has no intrinsic localisation signals and is normally surface expressed in a non-polarised manner (*Garrido, 2001*; *Fache et al., 2004*). We expressed chimeras of CD4 alone or CD4 fused to either ctCB1R$^{WT}$ or a ctCB1R lacking the *H9* domain (ctCB1R$^{\Delta H9}$; *Figure 3B*) in hippocampal neurons and examined surface expression by immunostaining (*Figure 3C*).

Analysis of the axon to dendrite ratio of surface expression (the surface polarity index) revealed that CD4-ctCB1R$^{WT}$ was markedly more axonally polarised than CD4 alone, indicating that ctCB1R may play a role in polarisation despite its lack of defined canonical localisation signals. Moreover, although still significantly axonally polarised, the degree of polarisation was significantly lower for CD4-ctCB1R$^{\Delta H9}$, suggesting that *H9* may also contribute to this process (*Figure 3D*).

We next analysed constitutive endocytosis of CD4-ctCB1R$^{WT}$ and CD4-ctCB1R$^{\Delta H9}$ (*Figure 4A,B*). *H9* does not determine surface polarity by driving differential constitutive endocytosis from either the dendritic or axonal membrane since there was no difference between the internalisation of CD4-ctCB1R$^{WT}$ or CD4-ctCB1R$^{\Delta H9}$. Interestingly, both CD4-ctCB1R$^{WT}$ and CD4-ctCB1R$^{\Delta H9}$ were

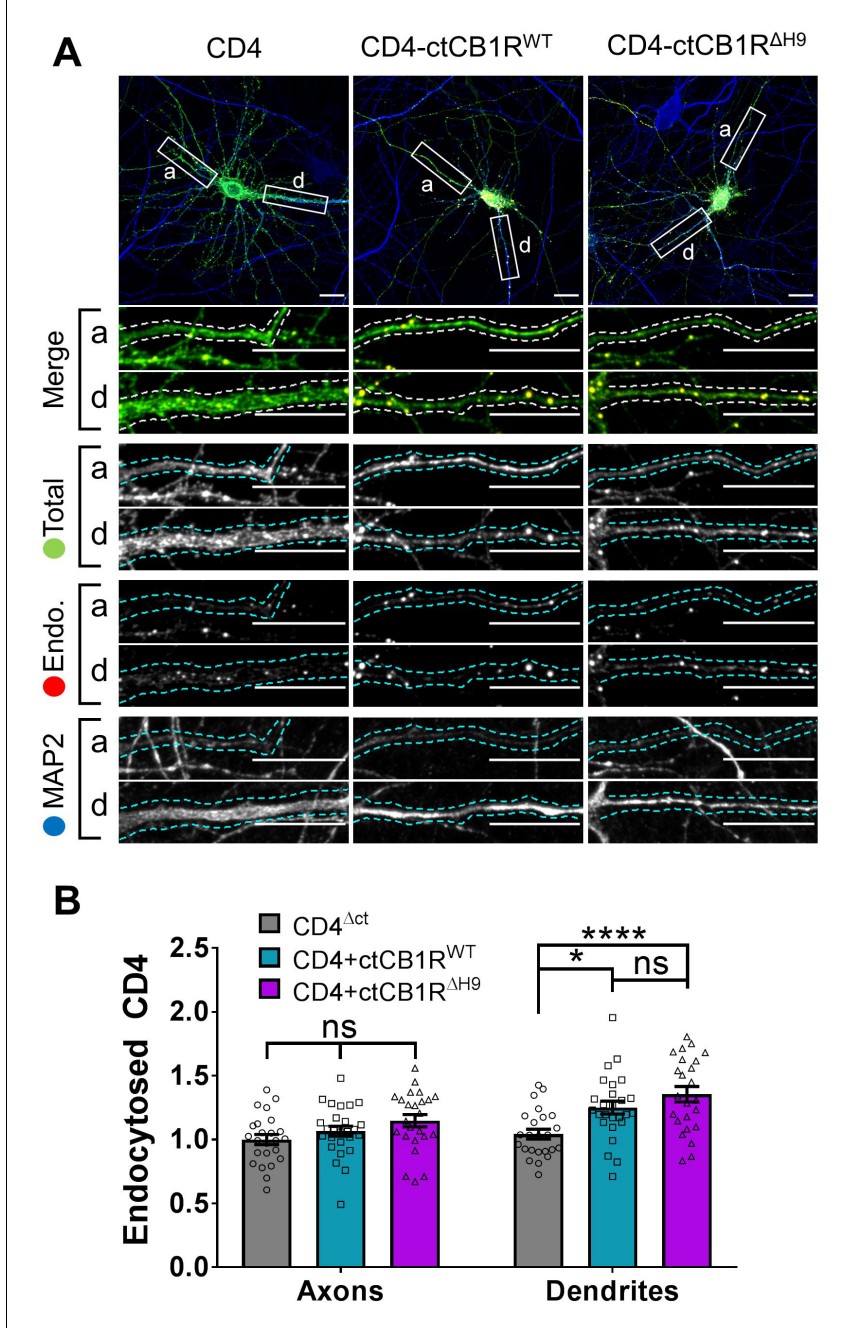

**Figure 4.** The C-terminal domain of CB1R plays a role in internalisation in dendrites independent of *H9*. (**A**) Representative confocal images of DIV 14 primary hippocampal neurons expressing CD4$^{\Delta CT}$ (left), CD4-ctCB1R$^{WT}$ (middle), or CD4-ctCB1R$^{\Delta H9}$ (right). Neurons were subjected to 2 hr of antibody feeding followed by stripping of surface antibody to reveal the endocytosed pool of receptors. Upper panels for each condition show a whole cell field of view and lower panels are enlargements of axonal (a) and dendritic (d) ROIs. Green = total; red = endocytosed; blue = dendritic marker (MAP2). Merge: endocytosed to total seen as yellow. Scale bar = 20 μm. (**B**) Quantification of data presented in (**A**). Both CD4-ctCB1R$^{WT}$ and CD4-ctCB1R$^{\Delta H9}$ were significantly more internalised in dendrites, but not in axons, than CD4 alone. Three independent experiments; n = 24–26 neurons per condition. Two way ANOVA with Sidak's *post hoc* test. Axons, CD4 vs. WT vs. ΔH9: mean ± SEM, 1.00 ± 0.040 vs. 1.065 ± 0.039 vs. 1.148 ± 0.048; N = 3, n = 24 vs. N = 3, n = 26 vs. N = 3, n = 24; $^{ns}p \geq 0.3514$. Dendrites, CD4 vs. WT: mean ± SEM, 1.042 ± 0.038 vs. 1.250 ± 0.051; N = 3, n = 24 vs. N = 3, n = 26; $*p = 0.0279$. Dendrites, CD4 vs. ΔH9: mean ± SEM, 1.042 ± 0.038 vs. 1.355 ± 0.060; N = 3, n = 24 vs. N = 3, n = 24; $*p < 0.0001$. Dendrites, WT vs. ΔH9: mean ± SEM, 1.250 ± 0.051 vs. 1.355 ± 0.060; N = 3, n = 26 vs. N = 3, n = 24; $^{ns}p = 0.8275$.
DOI: https://doi.org/10.7554/eLife.44252.006

significantly more internalised than CD4 alone in dendrites, but not in axons (*Figure 4A,B*). These results suggest that a domain other than *H9* promotes constitutive, but reportedly not activity-dependent (*McDonald et al., 2007a*), internalisation in dendrites. Importantly, however, because this increase in internalisation in identical between CD4-ctCB1R$^{WT}$ and ctCB1R$^{\Delta H9}$, this endocytic mechanism does not account for the failure of CD4-ctCB1R$^{\Delta H9}$ to polarise to the level of CD4-ctCB1R$^{WT}$.

### *H9* restricts delivery of CB1R to the dendritic membrane

To further explore the possibility that *H9* is involved in the axonal surface polarity of CB1R, we used RUSH to compare the forward trafficking of SBP-EGFP-CB1R$^{WT}$ and SBP-EGFP-CB1R$^{\Delta H9}$. We labelled all the CB1R that had been surface expressed (surface + endocytosed) 0, 30, 60, and 90 min after biotin-mediated release from the ER. Representative neuronal images at 90 min after biotin-mediated release are shown in *Figure 5A*.

Interestingly, significantly more SBP-EGFP-CB1R$^{\Delta H9}$ than SBP-EGFP-CB1R$^{WT}$ reached the surface of dendrites during the time course of our experiments (*Figure 5B*), whereas trafficking to axons was similar for both SBP-EGFP-CB1R$^{WT}$ and SBP-EGFP-CB1R$^{\Delta H9}$ (*Figure 5C*). These altered properties resulted in a significant difference in the surface + endocytosed polarity index after 90 min (*Figure 5D*) and are consistent with a role for *H9* in restricting delivery of CB1R to the dendritic membrane.

### *H9* plays a role in the surface retention of CB1R

Surprisingly, in contrast to the total amount of CB1R that had been surface expressed during the time course (surface + endocytosed; *Figure 5D*), the polarity of the amount of CB1R on the cell surface 90 min after biotin-mediated release was identical for SBP-EGFP-CB1R$^{WT}$ and SBP-EGFP-CB1R$^{\Delta H9}$ (surface; *Figure 5E*). Closer analysis revealed identical levels of axonal surface expression of both SBP-EGFP-CB1R$^{WT}$ and SBP-EGFP-CB1R$^{\Delta H9}$ 60 min after release from the ER. However, at 90 min there is significantly less surface expression of $\Delta H9$ mutant (*Figure 5F*) suggesting that, although similar amounts of SBP-EGFP-CB1R$^{WT}$ and SBP-EGFP-CB1R$^{\Delta H9}$ reach the surface, surface expression of SBP-EGFP-CB1R$^{\Delta H9}$ is less stable than that of the wild-type. Furthermore, in dendrites, the increased delivery and surface trafficking of the $\Delta H9$ mutant is counteracted by the fact that less is retained at the surface 60 min after ER release (*Figure 5G*). Taken together these results suggest that, separate from its role in restricting delivery to the dendritic membrane, *H9* also plays a role in membrane stability and retention at both axons and dendrites.

### *H9* stabilises CB1R at the surface

To investigate the role of *H9* in membrane stability, we next compared surface expression (*Figure 6A*) and endocytosis (*Figure 6B*) of EGFP-CB1R$^{WT}$ and EGFP-CB1R$^{\Delta H9}$ in axons and dendrites at steady-state. EGFP-CB1R$^{\Delta H9}$ displayed lower levels of surface expression (*Figure 6C*), as well as increased endocytosis (*Figure 6D*) in both axons and dendrites compared to EGFP-CB1R$^{WT}$, suggesting *H9* plays a role in stabilising CB1R at the surface of both axons and dendrites. Moreover, similar to our findings using RUSH, there was no difference in surface polarity between EGFP-CB1R$^{WT}$ and EGFP-CB1R$^{\Delta H9}$ (*Figure 6E*). These results suggest that, while *H9* plays a role in CB1R surface expression and endocytosis, its potential to mediate surface polarity is masked in the full-length receptor.

### CB1R$^{\Delta H9}$ is less efficient at activating downstream signalling pathways and more susceptible to agonist-induced internalisation

Because CB1R surface expression and polarisation has been linked to its activity (*Leterrier et al., 2006*; *Ladarre et al., 2014*), we investigated if deleting *H9* affects CB1R downstream signalling pathways. Measuring the signalling efficiency of EGFP-CB1R$^{\Delta H9}$ in neurons would require the complete removal of endogenous CB1R so we expressed EGFP-CB1R$^{WT}$ or EGFP-CB1R$^{\Delta H9}$ in HEK293T cells, which do not express endogenous CB1R (*Atwood et al., 2011*) and are routinely used to measure activation of the ERK pathway. Cells were treated with vehicle (EtOH) or stimulated with the selective CB1R agonist ACEA (arachidonyl-2'-chloroethylamide) (*Hillard et al., 1999*) and blotted for ERK1/2 phosphorylation as a measure of signalling downstream of CB1R (*Daigle et al., 2008*). There

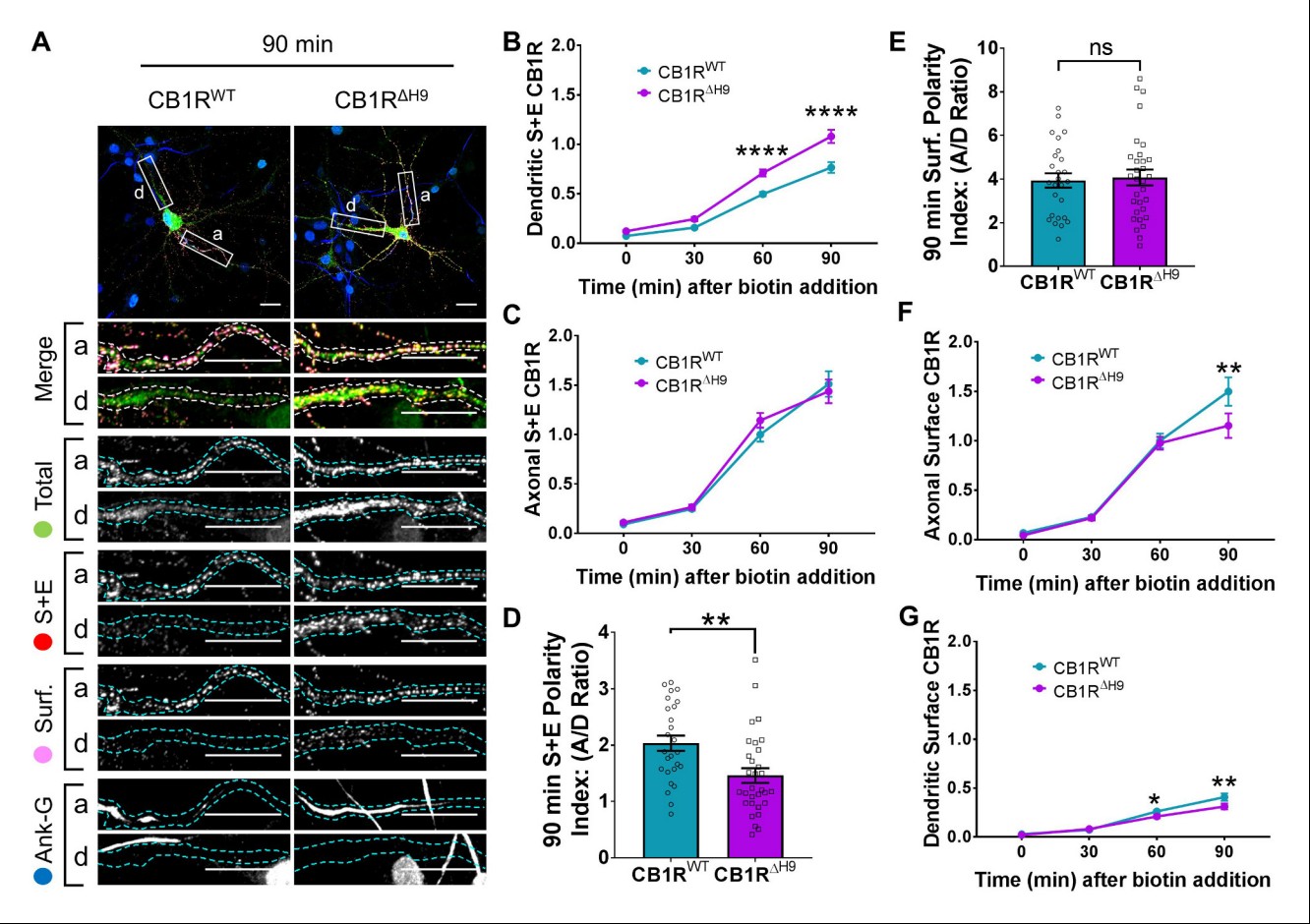

**Figure 5.** *H9* both restricts delivery of CB1R to the dendritic membrane and plays a role in surface retention of CB1R. The trafficking of RUSH SBP-EGFP-CB1R following release with biotin was monitored after 0 (no biotin), 30, 60, and 90 min in DIV 13 hippocampal neurons. (A) Representative confocal images of hippocampal neurons expressing SBP-EGFP-CB1R^WT or SBP-EGFP-CB1R^ΔH9 90 min after release with biotin. Upper panels for each condition show whole cell field of view and lower panels are enlargements of axonal (a) and dendritic (d) ROIs. Green = total; red = surface + endocytosed; magenta = surface; blue = axon marker (Ankyrin-G). Merge: surface to total seen as white; endocytosed to total seen as yellow. Scale bar = 20 µm. (B) Quantification of data represented in (A). Time-resolved analysis of surface + endocytosed receptors shows significantly more SBP-EGFP-CB1R^ΔH9 reaches the surface of dendrites than SBP-EGFP-CB1R^WT, indicating that *H9* may play a role in restricting delivery to the dendritic surface. Two-way ANOVA with Sidak's *post hoc* test. Three to seven independent experiments, n = 26–63 neurons per condition. 60 min, WT vs. ΔH9: mean ± SEM, 0.497 ± 0.022 vs. 0.711 ± 0.036; N = 8, n = 63 vs. N = 8, n = 48; ****p < 0.0001. 90 min, WT vs. ΔH9: mean ± SEM, 0.766 ± 0.054 vs. 1.08 ± 0.066; N = 4, n = 26 vs. N = 4, n = 31; ****p < 0.0001. C) Quantification of data represented in (A). Time-resolved analysis of surface + endocytosed receptors shows no difference between SBP-EGFP-CB1R^WT and SBP-EGFP-CB1^ΔH9 in reaching the surface of the axon. Two-way ANOVA with Sidak's *post hoc* test. N = three to seven independent experiments, n = 26–63 neurons per condition. 0, 30, 60, 90 min, WT vs. ΔH9: p > 0.2459. (D) Quantification of data represented in (A). Analysis of surface + endocytosed polarity demonstrates a defect in polarised delivery of SBP-EGFP-CB1R^ΔH9 compared to SBP-EGFP-CB1R^WT. Unpaired t-test. N = four independent experiments, n = 26–31 neurons per condition. WT vs. ΔH9: mean ± SEM, 2.03 ± 0.136 vs. 1.46 ± 0.13; N = 4, n = 26 vs. N = 4, n = 31; **p = 0.0038. (E) Quantification of data represented in (A). Analysis of surface polarity revealed no difference between SBP-EGFP-CB1R^WT and SBP-EGFP-CB1^ΔH9. Unpaired t-test. N = four independent experiments, n = 26–31 neurons per condition. WT vs. ΔH9: mean ± SEM, 3.935 ± 0.329 vs. 4.075 ± 0.361; N = 4, n = 26 vs. N = 4, n = 31; ^ns p = 0.7797. (F) Quantification of data represented in (A). Time-resolved analysis of surface receptors shows significantly less SBP-EGFP-CB1^ΔH9 than SBP-EGFP-CB1R^WT on the surface of axons 90 min after release, most likely due to increased endocytosis of the ΔH9 mutant. Two-way ANOVA with Sidak's *post hoc* test. N = three to eight independent experiments, n = 26–63 neurons per condition. 90, WT vs. ΔH9: mean ± SEM, 1.498 ± 0.144 vs. 1.154 ± 0.123; N = 4, n = 26 vs. N = 4, n = 31; **p = 0.0066. (G) Quantification of data represented in (A). Time-resolved analysis of surface receptors shows significantly less SBP-EGFP-CB1^ΔH9 than SBP-EGFP-CB1R^WT on the surface of dendrites 60 and 90 min after release, most likely due to increased endocytosis. Two-way ANOVA with Sidak's *post hoc* test. N = three to eight independent experiments, n = 26–63 neurons per condition. 60, WT vs. ΔH9: mean ± SEM, 0.262 ± 0.013 vs. 0.21 ± 0.018; N = 8, n = 63 vs. N = 8, n = 48; *p = 0.0232. 90, WT vs. ΔH9: mean ± SEM, 0.408 ± 0.038 vs. 0.312 ± 0.030; N = 4, n = 26 vs. N = 4, n = 31; **p = 0.0011.

DOI: https://doi.org/10.7554/eLife.44252.007

was no significant difference in ERK1/2 phosphorylation in cells expressing EGFP-CB1R$^{WT}$ or EGFP-CB1R$^{\Delta H9}$ under basal conditions in the absence of ACEA. However, upon ACEA stimulation, the level of ERK1/2 activation was significantly reduced in EGFP-CB1R$^{\Delta H9}$-transfected cells compared to EGFP-CB1R$^{WT}$-transfected cells expressing equivalent amounts of receptor (*Figure 7A–C*). Because the *ΔH9* mutant is more internalised than the wild-type in neurons, we examined whether the deficiency in ERK1/2 phosphorylation was due to a similarly reduced surface expression in HEK283T cells. However, EGFP-CB1R$^{WT}$ and EGFP-CB1R$^{\Delta H9}$ were surface expressed at equivalent levels in HEK293T cells, as determined by surface biotinylation experiments (*Figure 7D–E*), suggesting the *ΔH9* mutant is deficient in its ability to activate downstream signalling pathways.

We next monitored ACEA-induced internalisation of EGFP-CB1R$^{WT}$ and EGFP-CB1R$^{\Delta H9}$ in axons of hippocampal neurons (*Figure 7F*). ACEA-induced internalisation of EGFP-CB1R$^{\Delta H9}$ was significantly greater than that observed for EGFP-CB1R$^{WT}$ (*Figure 7F,G*). Taken together, these data indicate that CB1R$^{\Delta H9}$ is less stable at the axonal surface under basal conditions and that it is more susceptible to agonist-induced internalisation.

### The role of *H9* in polarity is revealed in the presence of inverse agonist

Our data thus far indicate that ctCB1R, and the *H9* domain in particular, can mediate surface polarity of a CD4 chimera (*Figure 3*), and promote polarised surface delivery of CB1R (*Figure 5*). In contrast, deletion of *H9* has no effect on CB1R surface polarity at steady-state (*Figure 6*). However, deletion of *H9* does have a striking effect on the surface stability of CB1R with CB1R$^{\Delta H9}$ being less surface expressed in both axons and dendrites and displaying increased endocytosis (*Figures 4* and *6*). Furthermore, CB1R $^{\Delta H9}$ is more responsive to agonist-induced internalisation (*Figure 7*).

We therefore wondered whether the difference between the CD4 chimeras and the full-length receptor and between surface + endocytosed and surface polarity may be attributable to the agonist binding capability of the full-length receptor. To test this we used the CB1R-specific inverse agonist AM281 to prevent the receptor entering an active conformation and which has previously been shown to increase somatodendritic surface expression similar to treatment with an endocytosis inhibitor (*Leterrier et al., 2006*). We reasoned that AM281 might reveal a difference in surface polarity between EGFP-CB1R$^{WT}$ and EGFP-CB1R$^{\Delta H9}$, like that observed with the CD4 chimeras and in surface + endocytosed polarity.

In hippocampal neurons treated with the DMSO control both EGFP-CB1R$^{WT}$ and EGFP-CB1R$^{\Delta H9}$ displayed similar levels of surface polarity (*Figure 8B*). In the presence of AM281, however, EGFP-CB1R$^{\Delta H9}$ had significantly reduced surface polarity compared EGFP-CB1R$^{WT}$ (*Figure 8B*) due to a significantly increased amount of dendritic surface expression (*Figure 8C*).

These results indicate that in the absence of constitutive activity of the receptor, *H9* plays a role in mediating CB1R surface polarity. Furthermore, these data suggest that the increased internalisation observed in dendrites with *H9* deletion may be mediated by the presence of endogenous agonist and reaffirm the importance of the neuronal milieu on CB1R trafficking.

### Discussion

Consistent with previous reports (*Leterrier et al., 2006*; *Simon et al., 2013*; *Rozenfeld and Devi, 2008*; *McDonald et al., 2007b*) we observed highly axonally polarised surface expression of CB1R. As illustrated in schematic form in *Figure 9*, our data indicate that two distinct, but complementary, mechanisms contribute to this polarisation of CB1Rs. *1)* Using time-resolved RUSH assays we demonstrate that more *de novo* CB1R is delivered to the axon and that it is more rapidly surface expressed than in dendrites. *2)* Once at the axonal membrane the newly delivered CB1R is more stably retained whereas in dendritic membrane CB1R surface expression is transient and it is rapidly internalised. It should be noted, however, that our data do not exclude additional possibilities, including that CB1R internalised in the somatodendritic compartment can be rerouted to the axon via the transcytosis pathway, thus further facilitating axonal polarity (*Simon et al., 2013*). Furthermore, since CD4-ctCB1R$^{WT}$ and CD4-ctCB1R$^{\Delta H9}$ chimeras cannot bind agonist, our results are consistent with ctCB1R contributing to constitutive polarisation via a mechanism distinct from the proposed continuous activation of CB1R by the presence of the endogenous agonist 2-Arachidonoylglycerol (2-AG) in the dendritic membrane (*Ladarre et al., 2014*). Thus, we conclude that ctCB1R contributes to the constitutive preferential delivery of CB1R to the axonal membrane.

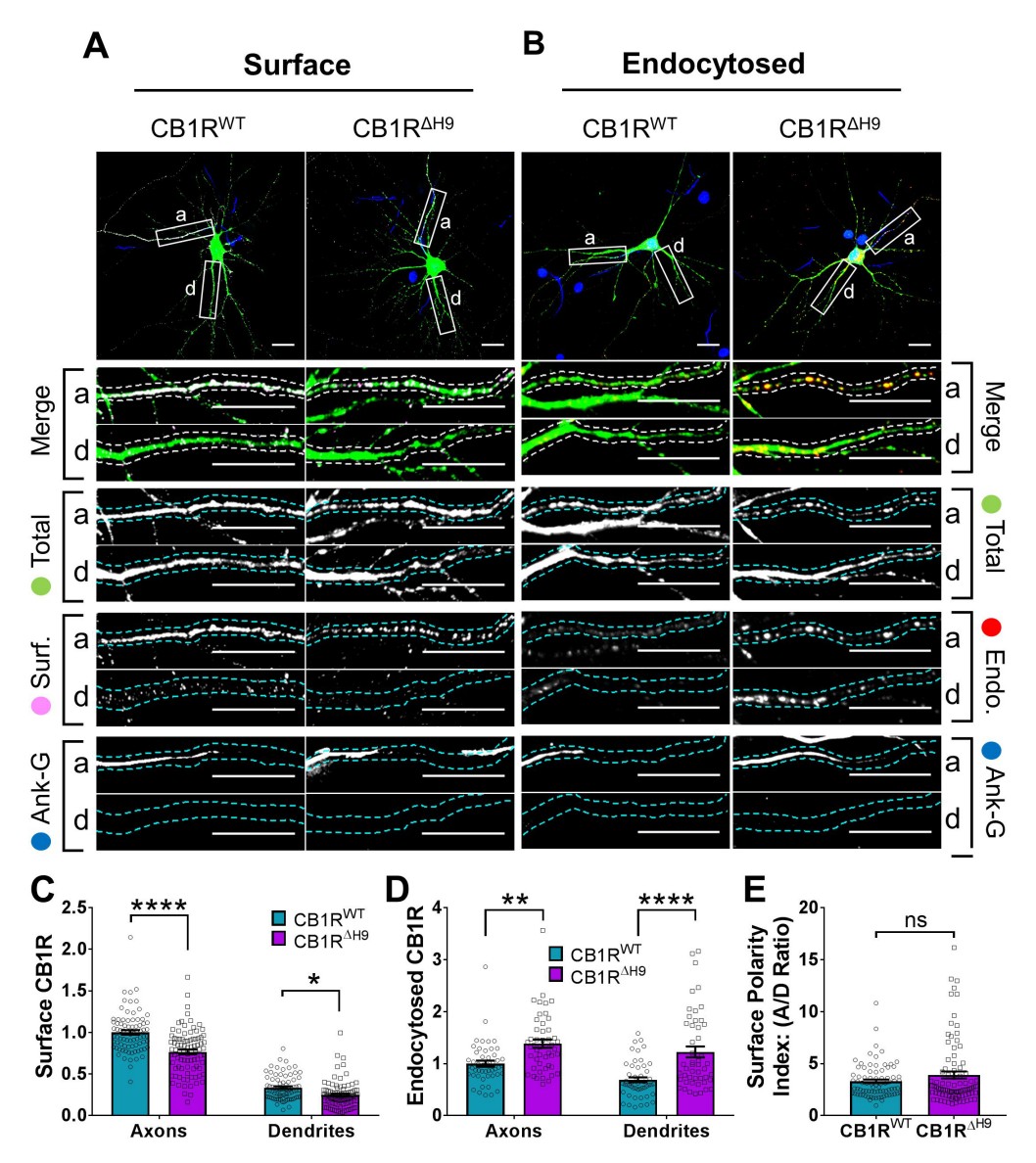

**Figure 6.** *H9* stabilises CB1R at the axonal surface. (**A**) Representative confocal images of surface stained DIV 14 hippocampal neurons expressing EGFP-CB1R$^{WT}$ or EGFP-CB1R$^{ΔH9}$. Green = total; magenta = surface; blue = axon marker (Ankyrin-G). Merge: surface to total seen as white. (**B**) Representative confocal images of DIV 14 primary hippocampal neurons expressing EGFP-CB1R$^{WT}$ or EGFP-CB1R$^{ΔH9}$. Neurons were subjected to 2 hr of antibody feeding followed by stripping off of surface antibody to reveal the endocytosed pool of receptors. Green = total; red = endocytosed; blue = axon marker (Ankyrin-G). Merge: endocytosed to total seen as yellow. (**C**) Quantification of data shown in (**A**). Surface expression of EGFP-CB1R$^{ΔH9}$ in both axons and dendrites was significantly reduced compared to EGFP-CB1R$^{WT}$. Two-way ANOVA with Tukey's *post hoc* test. N = ten independent experiments; n = 80–88 neurons per condition. Axons, WT vs. ΔH9: mean ± SEM, 1 ± 0.028 vs. 0.765 ± 0.029; N = 10, n = 80 vs. N = 10, n = 88; ****p < 0.0001. Dendrites, WT vs. ΔH9: mean ± SEM, 0.335 ± 0.016 vs. 0.247 ± 0.017; N = 10, n = 80 vs. N = 10, n = 88; *p = 0.0392. (**D**) Quantification of data shown in (**B**). Endocytosis of EGFP-CB1R$^{ΔH9}$ is significantly increased compared to EGFP-CB1R$^{WT}$ in both axons and dendrites. One-way ANOVA with Tukey's *post hoc* test. N = seven independent experiments; n = 49 neurons per condition. Axons, WT vs. ΔH9: mean ± SEM, 1.00 ± 0.058 vs. 1.38 ± 0.08; **p = 0.0026. Dendrites, WT vs. ΔH9: mean ± SEM, 0.689 ± 0.05 vs. 1.225 ± 0.105; ****p < 0.0001. (**E**) Quantification of data shown in (**A**) presented as the surface polarity index. There was no difference in surface polarity between EGFP-CB1R$^{WT}$ or EGFP-CB1R$^{ΔH9}$. Mann-Whitney test. N = ten independent experiments; n = 80–88 neurons per condition. WT vs. ΔH9: mean ± SEM, 3.298 ± 0.1812 vs. 3.915 ± 0.3367; N = 10, n = 80 vs. N = 10, n = 88; p = 0.6886.
DOI: https://doi.org/10.7554/eLife.44252.008

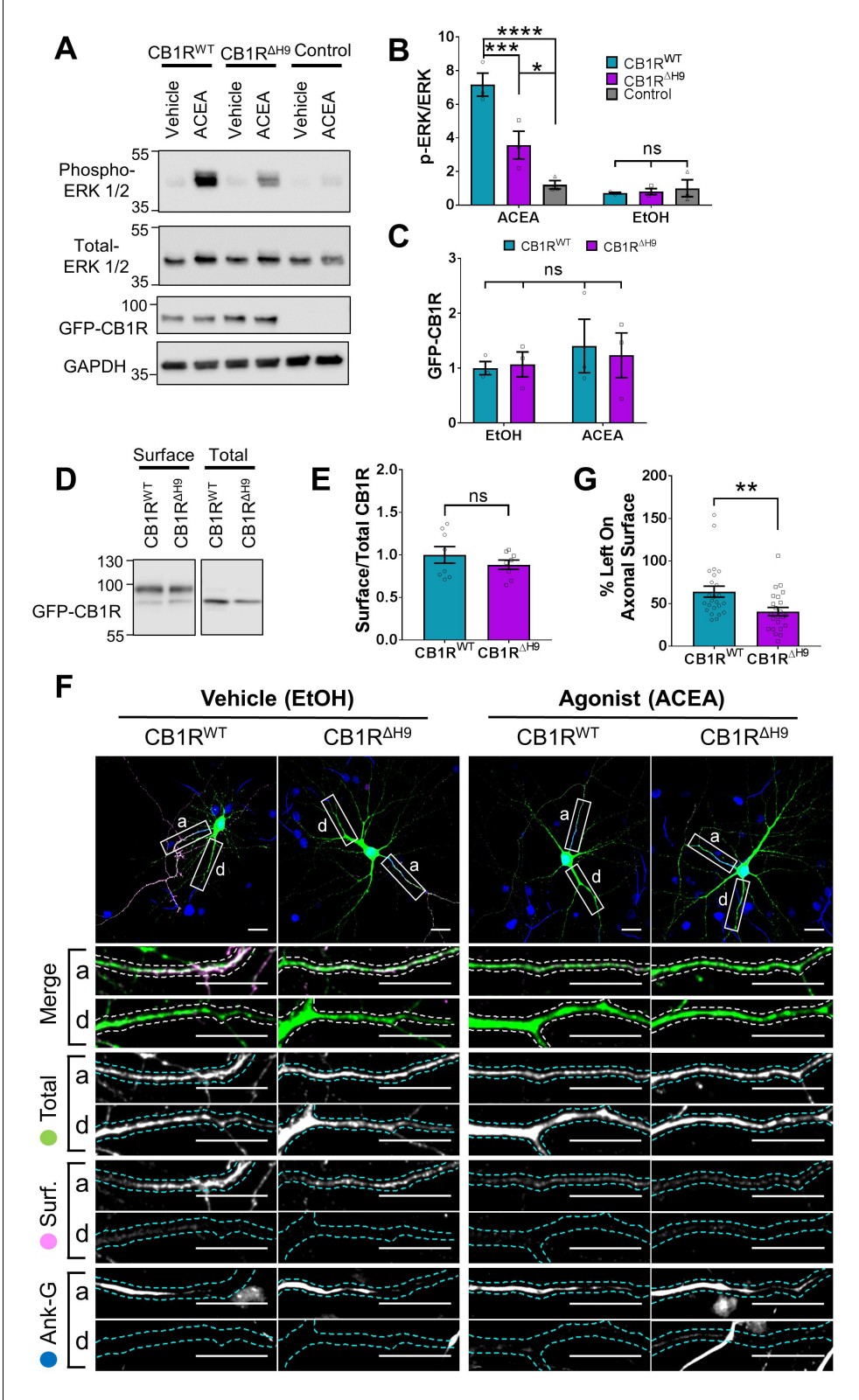

**Figure 7.** Role of *H9* in CB1R signalling and in resisting agonist-induced endocytosis. (**A**) Representative blots showing ERK1/2 phosphorylation in HEK293T cells expressing EGFP-CB1R^WT or EGFP-CB1R^ΔH9 following vehicle (0.1% EtOH) or ACEA (1 µM) treatment for 5 min. (**B**) Quantification of data shown in (**A**). Following treatment with ACEA, ERK1/2 was significantly more phosphorylated in EGFP-CB1R^WT- and EGFP-CB1R^ΔH9-transfected cells

*Figure 7 continued on next page*

*Figure 7 continued*

compared to untransfected cells. However, ERK1/2 activation was significantly reduced in EGFP-CB1R$^{\Delta H9}$-expressing cells compared to EGFP-CB1R$^{WT}$-expressing cells. There was no significant difference in ERK1/2 phosphorylation in vehicle-treated cells. Two-way ANOVA with Tukey's *post hoc* test. N = three independent experiments. ACEA, WT vs. Control: mean ± SEM, 7.17 ± 0.684 vs. 1.21 ± 0.252; ****p < 0.0001. ΔH9 vs. Control: mean ± SEM, 3.57 ± 0.825 vs. 1.21 ± 0.252; *p = 0.0150. WT vs. ΔH9: mean ± SEM, 7.17 ± 0.684 vs. 3.57 ± 0.825; ***p = 0.0007. EtOH, WT vs. ΔH9 vs. Control: $^{ns}$p ≥ 0.9125. (**C**) Quantification of data shown in (**A**). EGFP-CB1R$^{WT}$ and EGFP-CB1R$^{\Delta H9}$ expressed equally in HEK293T cells. Two-way ANOVA with Sidak's *post hoc* test. Three independent experiments. ($^{ns}$p ≥ 0.9654). (**D**) Representative immunoblots from surface biotinylation experiments showing surface and total fractions of EGFP-CB1R$^{WT}$ and EGFP-CB1R$^{\Delta H9}$ in HEK293T cells. (**E**) Quantification of data shown in (**D**). EGFP-CB1R$^{WT}$ and EGFP-CB1R$^{\Delta H9}$ are surface expressed at equivalent levels in HEK293T cells. Unpaired t-test. Eight independent experiments. WT vs. ΔH9: mean ± SEM, 1.00 ± 0.0974 vs. 0.885 ± 0.0549; $^{ns}$p = 0.3212. (**F**) Representative confocal images of DIV 12 hippocampal neurons expressing EGFP-CB1R$^{WT}$ or EGFP-CB1R$^{\Delta H9}$ and treated with vehicle (0.1% EtOH) or CB1R agonist (5 μM ACEA) for 3 hr. Upper panels for each condition show whole cell field of view and lower panels are enlargements of axonal (**a**) and dendritic (**d**) ROIs. Green = total; magenta = surface; blue = axon marker (Ankyrin-G). Merge: surface to total seen as white. (**G**) Quantification of data represented in (**F**). Significantly less EGFP-CB1R$^{\Delta H9}$ than EGFP-CB1R$^{WT}$ remained on the surface of axons after agonist application, indicating greater sensitivity to agonist-induced internalisation. The surface mean fluorescence was first normalised to the total mean fluorescence for each ROI, then to the average axonal EtOH value within a condition (set to 100%). Unpaired t-test. N = three independent experiments; n = 23–24 neurons per condition. WT vs. ΔH9: mean ± SEM, 64 ± 6.42 vs. 40.6 ± 4.87; N = 3, n = 24 vs. N = 3, n = 23; **p = 0.0059.

DOI: https://doi.org/10.7554/eLife.44252.009

Our results further demonstrate that ctCB1R is important for maintaining axonal surface polarity, in part mediated by the *H9* motif, which plays a role in both the preferential delivery and selective retention of CB1R at in axons. We show that deleting *H9* (CB1R$^{\Delta H9}$) has a range of effects on trafficking, surface expression, and signalling in hippocampal neurons. More specifically, these include; *i*) CB1R$^{\Delta H9}$ lacks the preferential delivery to axons observed for CB1R$^{WT}$, *ii*) CB1R$^{\Delta H9}$ is less efficiently surface expressed, *iii*) CB1R$^{\Delta H9}$ that does reach the surface it is more rapidly endocytosed in both axons and dendrites, and *iv*) CB1R$^{\Delta H9}$ is more sensitive to agonist-induced internalisation and less efficient at downstream signalling, monitored by activation of ERK1/2 phosphorylation.

## Preferential axonal trafficking

The mechanism behind polarised membrane trafficking in neurons is a fundamental question and our data suggest a sorting mechanism at the level of the secretory pathway that preferentially targets CB1R to the axon. Since dendritic and axonal cargo are synthesized in the somatodendritic compartment, selective sorting to the correct domain is crucial. While several sorting signals and adaptors have been described for dendritic cargo, the mechanisms behind selective sorting to axons are less well known (*Lasiecka and Winckler, 2011*; *Bentley and Banker, 2016*). For example, a recent study in *C. elegans* has suggested that sorting of cargos to axons or dendrites depends on binding to different types of clathrin-associated adaptor proteins (AP); axonal cargo bind to AP-3 whereas dendritic cargo bind to AP-1 (*Li et al., 2016*). Interestingly, AP-3 binding has been associated with CB1R trafficking to the lysosome in the dendritic compartment (*Rozenfeld and Devi, 2008*). One possibility is that *H9* may modulate CB1R binding to AP-3, allowing for preferential delivery to axons and sorting to dendritic lysosomes, causing an decrease in dendritic membrane CB1R. More studies are needed to examine the possibility of *H9* influencing AP-3 and CB1R interaction.

## Trafficking within the axon

We used time-resolved RUSH experiments to investigate the initiation of CB1R polarity. We measured the transit through the secretory pathway and incorporation into, and passage through, the highly organised axon initial segment (AIS) that acts as a 'gate-keeper' for proteins entering the axonal compartment. Our data show that CB1R polarisation was initiated in the first 90 min since they were directly targeted to, and surface expressed within, proximal axonal regions.

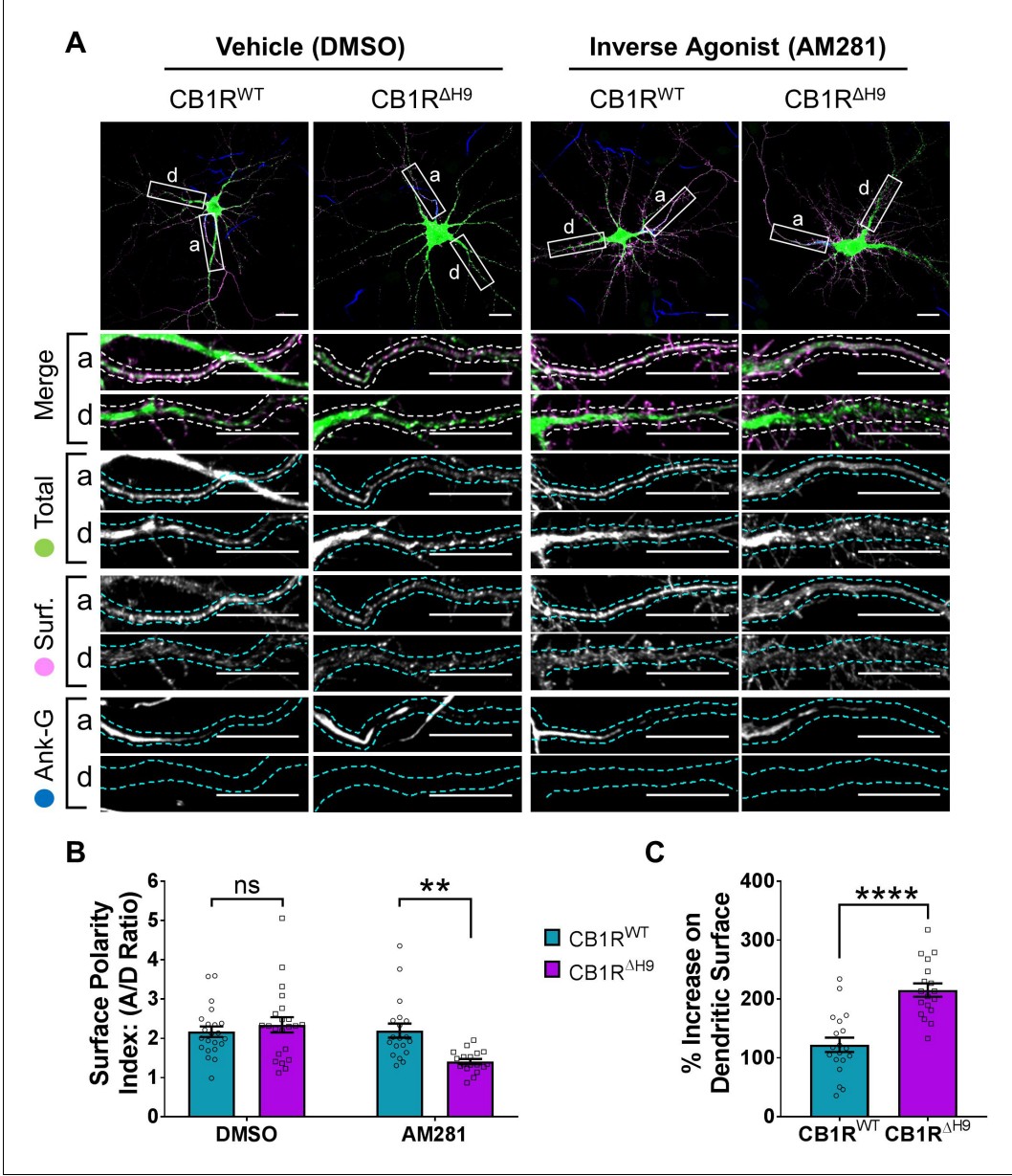

**Figure 8.** The role of *H9* in polarity is revealed in the presence of inverse agonist. (**A**) Representative confocal images of DIV 14 hippocampal neurons expressing EGFP-CB1R$^{WT}$ or EGFP-CB1R$^{\Delta H9}$ and treated with vehicle (0.2% DMSO) or CB1R inverse agonist (10 μM AM281) for 3 hr. Upper panels for each condition show whole cell field of view and lower panels are enlargements of axonal (**a**) and dendritic (**d**) ROIs. Green = total; magenta = surface; blue = axon marker (Ankyrin-G). Merge: surface to total seen as white. (**B**) Quantification of data shown in (**A**) presented as the surface polarity index (A/D ratio). In the presence of inverse agonist, but not vehicle, EGFP-CB1R$^{\Delta H9}$ was significantly less axonally polarised than EGFP-CB1R$^{WT}$. Two-way ANOVA with Sidak's *post hoc* test. N = three independent experiments; n = 18–22 neurons per condition. DMSO, WT vs. ΔH9: mean ± SEM, 2.17 ± 0.135 vs. 2.34 ± 0.196; N = 3, n = 22 vs. N = 3, n = 22; $^{ns}$p = 0.9605. AM281, WT vs. ΔH9: mean ± SEM, 2.2 ± 0.18 vs. 1.41 ± 0.0649; N = 3, n = 19 vs. N = 3, n = 18; **p = 0.0067. (**C**) Quantification of data represented in (**A**). Significantly more EGFP-CB1R$^{\Delta H9}$ than EGFP-CB1R$^{WT}$ relocated to the surface of dendrites after inverse agonist application. The surface mean fluorescence was first normalised to the total mean fluorescence for each ROI, then to the average DMSO value within a condition (set to 100%). Unpaired t-test. N = three independent experiments; n = 18–19 neurons per condition. WT vs. ΔH9: mean ± SEM, 122 ± 12.2 vs. 215 ± 11.3; N = 3, n = 19 vs. N = 3, n = 18; ****p < 0.0001.

DOI: https://doi.org/10.7554/eLife.44252.010

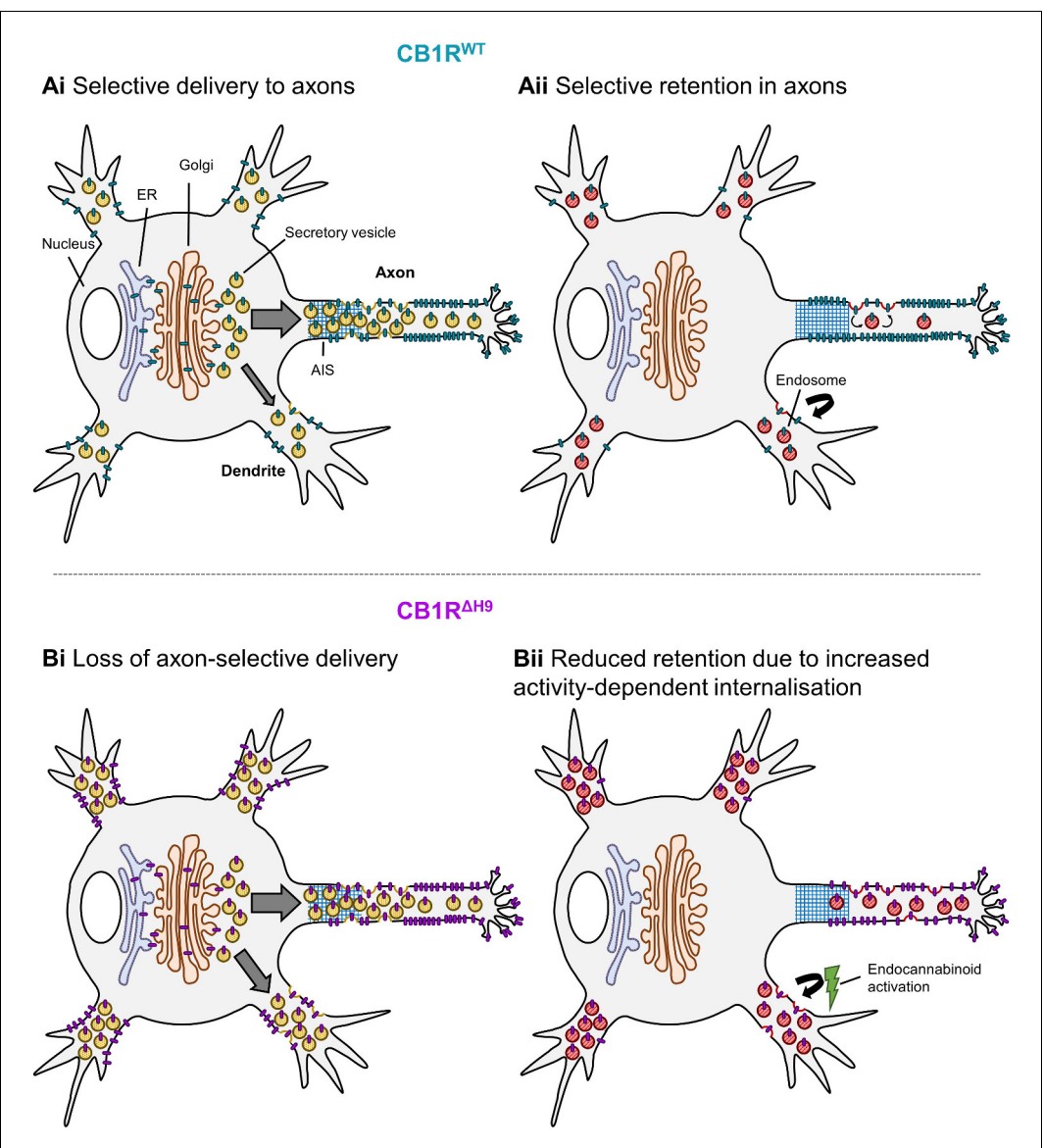

**Figure 9.** Schematic summarising main findings. Polarised surface distribution of CB1R is established and maintained by two complementary mechanisms: (**Ai**) selective delivery of newly synthesized CB1R to the axon (or restricted delivery to dendrites) and (**Aii**) selective retention in axons and retrieval from dendrites. The C-terminal motif Helix 9 plays a role in both of these mechanisms because deletion of *H9* leads to: (**Bi**) a loss of axon-selective delivery and (**Bii**) reduced retention in both axons and dendrites that can be reversed in dendrites by inverse agonist application, suggesting that CB1R$^{\Delta H9}$ is more susceptible to activity-driven internalisation.
DOI: https://doi.org/10.7554/eLife.44252.011

Immunocytochemistry in brain sections using immunogold electron microscopy (*Katona et al., 1999*; *Nyíri et al., 2005*) or STORM super-resolution imaging (*Dudok et al., 2015*) detect CB1R predominantly at the presynaptic terminal, consistent with a disto-proximal gradient of expression at the axonal plasma membrane (*Simon et al., 2013*). Therefore, given the highly branched morphology of typical CB1R expressing neurons, correct axonal polarization likely requires specific distal targeting mechanisms in addition to the processes we describe using time-resolved RUSH experiments.

CB1Rs are highly mobile and diffuse rapidly in the plasma membrane (*Mikasova et al., 2008*; *Oddi et al., 2012*), so the accumulation of surface CB1R we observe may be followed by lateral diffusion and 'capture' of surface CB1R at presynaptic sites, analogous to the diffusion and retention

models proposed for AMPARs and GABA$_A$Rs at the postsynaptic membrane (*Hastings and Man, 2018*; *Kneussel and Hausrat, 2016*). Another, non-exclusive possibility is that CB1R-containing vesicles originating from the secretory system and/or endosomal system traffic to distal sites inside the axon (*Lasiecka and Winckler, 2011*). Indeed, it has been reported that CB1R in somatodendritic endosomes can be rerouted and trafficked to distal axonal surfaces (*Simon et al., 2013*). Our observation that intracellular CB1R is present at least 100 µm along the axon before it appears at the surface supports the concept of a rapid and direct trafficking of CB1R-containing secretory vesicles to more distal areas of the axon, although more detailed tracking of these secretory vesicles to presynaptic boutons would be required to confirm this. Overall, we interpret our data to suggest that arrival at, and progression through, the AIS constitutes the initial phase of CB1R axonal polarisation. Once within the axonal compartment trafficking to more distal locations and to presynaptic sites is then mediated by additional mechanisms that probably include both intracellular transport and lateral diffusion and trapping.

### *H9* and membrane retention

Our data suggest that *H9* stabilises CB1R at the membrane, regardless of compartment. While the *H8* domain is highly conserved in GPCRs, structural domains analogous to *H9* have only been reported in squid rhodopsin (*Murakami and Kouyama, 2008*) and the bradykinin receptor (*Piserchio et al., 2005*). NMR and circular dichroism studies suggest that *H9*, like *H8*, is an amphipathic α-helix, associating with the lipid bilayer via a cluster of hydrophobic residues on the non-polar face of the helix (*Ahn et al., 2009*). Furthermore, CB1R has been shown to be palmitoylated just downstream of *H8* at C416, which affects its membrane association and G-protein coupling (*Oddi et al., 2012*; *Oddi et al., 2018*). *H9* also contains a cysteine residue, raising the possibility that post-translational modifications such as palmitoylation, prenylation, or farnesylation at this site could modulate membrane association.

Since our data suggest that *H9* stabilises CB1R at the membrane, it is possible that the membrane association of *H9* could mask internalisation signals or interacting motifs. Consistent with this possibility, ctCB1R interacting proteins regulate CB1R endocytosis. For example, SGIP1, a protein linked to clathrin-mediated endocytosis, binds at an as yet undefined site on ctCB1R downstream of *H8*, preventing internalisation of activated CB1R (*Hájková et al., 2016*). Similarly, cannabinoid receptor interacting protein 1a (CRIP1a), which interacts directly with a motif in the last 9 C-terminal amino acids (*Niehaus et al., 2007*), reduces constitutive CB1R internalisation (*Mascia et al., 2017*) by competing with β-Arrestin binding (*Blume et al., 2017*). Therefore, it is possible that *H9* mediates the interactions between CB1R and SGIP1 and/or selectively promotes β-Arrestin rather than CRIP1a binding. Further studies examining the interaction between CB1R$^{WT}$, CB1R$^{ΔH9}$, CRIP1a, β-Arrestin1/2, and SGIP1 are needed to examine the mechanism by which *H9* stabilises surface CB1R.

Given the increased interest in CB1R as a clinical target, understanding the fundamental cell biology and trafficking behaviour of CB1R is an increasingly active and important area of research. Taken together, our results reveal that the C-terminal domain, and *H9* in particular, play important roles in trafficking of CB1R. These findings provide important insight into the mechanisms of CB1R polarity and highlight *H9* as an important regulator of CB1R endocytosis and surface expression.

## Materials and methods

Key resources table

| Reagent type (species) or resource | Designation | Source or reference | Identifiers | Additional information |
|---|---|---|---|---|
| Antibody | anti-GFP (chicken polyclonal) | Abcam | Abcam:ab13970; RRID:AB_300798 | ICC (1:1,000) |
| Antibody | anti-GFP (rat monoclonal; clone 3H9) | Chromotek | Chromotek:3h9-100; RRID:AB_10773374 | WB (1:2,000-1:5,000) |
| Antibody | anti-Ankyrin-G (mouse monoclonal; purified; clone N106/36) | NeuroMab | NeuroMab:75–146; RRID:AB_10673030 | ICC (1:500) |

*Continued on next page*

*Continued*

| Reagent type (species) or resource | Designation | Source or reference | Identifiers | Additional information |
|---|---|---|---|---|
| Antibody | anti-CD4 (mouse monoclonal; purified; clone OKT4) | BioLegend | BioLegend: 317402; RRID:AB_571963 | ICC (1:400) |
| Antibody | anti-MAP2 (rabbit polyclonal) | Synaptic Systems | SySy:188 003; RRID:AB_2281442 | ICC (1:500) |
| Antibody | anti-MAP Kinase, Activated/ monophosphorylated, Phosphothreonine ERK-1 and 2 (mouse monoconal; clone ERK-PT115) | Merck (Sigma-Aldrich) | Sigma-Aldrich: M7802; RRID:AB_260658 | WB (1:1,000) |
| Antibody | anti-MAP Kinase, Non-Phosphorylated ERK (mouse monoclonal; clone ERK-NP2) | Merck (Sigma-Aldrich) | Sigma-Aldrich: M3807; RRID:AB_260501 | WB (1:250) |
| Antibody | anti-GAPDH (mouse monoclonal; clone 6C5) | Abcam | Abcam:ab8245; RRID:AB_2107448 | WB (1:20,000) |
| Antibody (secondary) | Peroxidase anti-mouse IgG (goat) | Merck (Sigma-Aldrich) | Sigma-Aldrich: A3682 | WB (1:10,000) |
| Antibody (secondary) | Peroxidase anti-rat IgG (rabbit) | Merck (Sigma-Aldrich) | Sigma-Aldrich: A5795 | WB (1:10,000) |
| Antibody (secondary) | Cy2 anti-chicken IgY (donkey) | Jackson Immuno Research (Stratech) | JIR:703-225-155; RRID:AB_2340370 | ICC (1:400) |
| Antibody (secondary) | Alexa Fluor 647 anti-chicken IgY (donkey) | Jackson Immuno Research (Stratech) | JIR:703-606-155; RRID:AB_2340380 | ICC (1:400) |
| Antibody (secondary) | Cy3 anti-chicken IgY (donkey) | Jackson Immuno Research (Stratech) | JIR:703-165-155; RRID:AB_2340363 | ICC (1:400) |
| Antibody (secondary) | Cy5 anti-mouse IgG (donkey) | Jackson Immuno Research (Stratech) | JIR:715-175-150; RRID:AB_2340819 | ICC (1:400) |
| Antibody (secondary) | Cy3 anti-mouse IgG (donkey) | Jackson Immuno Research (Stratech) | JIR:715-165-150; RRID:AB_2340813 | ICC (1:400) |
| Antibody (secondary) | DyLight 405 anti-mouse IgG (goat) | Jackson Immuno Research (Stratech) | JIR:115-475-003; RRID:AB_2338786 | ICC (1:400) |
| Antibody (secondary) | Alexa Fluor 488 anti-rabbit IgG (donkey) | Jackson Immuno Research (Stratech) | JIR:711-545-152; RRID:AB_2313584 | ICC (1:400) |
| Chemical compound, drug | Arachidonyl-2′-chloroethylamide (ACEA) | Bio-Techne (Tocris) | Tocris:1319 | |
| Chemical compound, drug | AM 281 | Bio-Techne (Tocris) | Tocris:1115 | Dissolved in DMSO |
| Cell line (*H. sapiens*) | HEK293T | ECACC (Sigma-Aldrich) | ECACC:12022001; RRID:CVCL_0063 | |
| Cell line (*R. norvegicus*) | primary hippocampal neurons | University of Bristol Animal Services Unit | | E18 Wistar Han rats |
| Recombinant DNA reagent | pIRESneo3_Str-KDEL_IRES-SP$^{II-2}$-SBP-mCherry-Ecadherin (plasmid) | PMID:22406856 | Addgene:65287 | |

*Continued*

| Reagent type (species) or resource | Designation | Source or reference | Identifiers | Additional information |
|---|---|---|---|---|
| Recombinant DNA reagent | pcDNA1-SP$^{HgH}$-SEP-CB1R$^{WT}$ (plasmid) | PMID:17467290 | | |
| Recombinant DNA reagent | pcDNA1-SP$^{HgH}$-SEP-CB1R$^{\Delta H9}$ (plasmid) | This paper | | PCR mutagenesis template: pcDNA1-SP$^{HgH}$-SEP-CB1R$^{WT}$ |
| Recombinant DNA reagent | pIRESneo3_Str-KDEL_IRES-SP$^{Il-2}$-SBP-EGFP-CB1R$^{WT}$ (plasmid) | This paper | | PCR template: pcDNA1-SP$^{HgH}$-SEP-CB1R Vector: pIRESneo3_Str-KDEL_IRES-SP$^{Il-2}$-SBP-mCherry-Ecadherin |
| Recombinant DNA reagent | pIRESneo3_Str-KDEL_IRES-SP$^{Il-2}$-SBP-EGFP-CB1R$^{\Delta H9}$ (plasmid) | This paper | | PCR template: pcDNA1-SP$^{HgH}$-SEP-CB1R$^{\Delta H9}$ Vector: pIRESneo3_Str-KDEL_IRES-SP$^{Il-2}$-SBP-mCherry-Ecadherin |
| Recombinant DNA reagent | pcDNA3.1-SP$^{Il-2}$-SBP-EGFP-CB1R$^{WT}$ (plasmid) | This paper | | PCR template: pIRESneo3_Str-KDEL_IRES-SP$^{Il-2}$-SBP-EGFP-CB1R$^{WT}$ Vector: pcDNA3.1(+) |
| Recombinant DNA reagent | pcDNA3.1-SP$^{Il-2}$-SBP-EGFP-CB1R$^{\Delta H9}$ (plasmid) | This paper | | PCR template: pIRESneo3_Str-KDEL_IRES-SP$^{Il-2}$-SBP-EGFP-CB1R$^{\Delta H9}$ Vector: pcDNA3.1(+) |
| Recombinant DNA reagent | pcDNA3.1-SP$^{Il-2}$-EGFP-CB1R$^{WT}$ (plasmid) | This paper | | PCR mutagenesis template: pcDNA3.1-SP$^{Il-2}$-SBP-EGFP-CB1R$^{WT}$ |
| Recombinant DNA reagent | pcDNA3.1-SP$^{Il-2}$-EGFP-CB1R$^{\Delta H9}$ (plasmid) | This paper | | PCR mutagenesis template: pcDNA3.1-SP$^{Il-2}$-SBP-EGFP-CB1R$^{\Delta H9}$ |
| Recombinant DNA reagent | pCB6-CD4$^{\Delta Ct}$ (plasmid) | PMID:11689435 | | |
| Recombinant DNA reagent | pCB6-CD4-ctCB1R$^{WT}$ (plasmid) | This paper | | Overlap extension PCR template: pcDNA1-SP$^{HgH}$-SEP-CB1R$^{WT}$ Vector: pCB6-CD4$^{\Delta Ct}$ |
| Recombinant DNA reagent | pCB6-CD4-ctCB1R$^{\Delta H9}$ (plasmid) | This paper | | Overlap extension PCR template: pcDNA1-SP$^{HgH}$-SEP-CB1R$^{\Delta H9}$ Vector: pCB6-CD4$^{\Delta Ct}$ |
| Chemical compound, drug | Neurobasal | Thermo Fisher Scientific (Gibco) | TFS:21103049 | |
| Chemical compound, drug | Horse Serum | Merck (Sigma-Aldrich) | Sigma-Aldrich: H1270 | |
| Chemical compound, drug | B27 | Thermo Fisher Scientific (Gibco) | TFS:A3582801 | |
| Chemical compound, drug | GS21 | MTI GlobalStem | GS:3100 | |

*Continued on next page*

*Continued*

| Reagent type (species) or resource | Designation | Source or reference | Identifiers | Additional information |
|---|---|---|---|---|
| Chemical compound, drug | GlutaMAX | Thermo Fisher Scientific (Gibco) | TFS:35050038 | |
| Chemical compound, drug | DMEM | Lonza | Lonza:12–614F | |
| Chemical compound, drug | L-Glutamine | Merck (Sigma-Aldrich) | Sigma-Aldrich: G7513 | |
| Chemical compound, drug | FBS | Merck (Sigma-Aldrich) | Sigma-Aldrich: F7524 | |
| Chemical compound, drug | Lipofectamine 2000 | Thermo Fisher Scientific | TFS:11668019 | |
| Chemical compound, drug | D-Biotin | Merck (Sigma-Aldrich) | Sigma-Aldrich: B4501 | |
| Chemical compound, drug | EZ-Link Sulfo-NHS-SS-Biotin | Thermo Fisher Scientific | TFS:21331 | |
| Chemical compound, drug | Streptavidin—Agarose | Merck (Millipore) | Merck:S1638 | |
| Chemical compound, drug | Bovine Serum Albumin | Thermo Fisher Scientific | TFS:BP9704 | |
| Chemical compound, drug | Fluoromount-G | Thermo Fisher Scientific | TFS:00-4958-02 | |
| Chemical compound, drug | cOmplete Protease Inhibitor Cocktail | Merck (Sigma-Aldrich) | Sigma-Aldrich: 11697498001 | |
| Chemical compound, drug | Pierce Phosphatase Inhibitor | Thermo Fisher Scientific | TFS:A32957 | |
| Chemical compound, drug | CHAPS Detergent | Thermo Fisher Scientific | TFS:28300 | |
| Software, algorithm | GraphPad Prism | GraphPad Prism (https://graphpad.com) | RRID:SCR_002798 | Version 7 |
| Software, algorithm | Fiji | Fiji (https://fiji.sc/) | RRID:SCR_002285 | |

## Constructs and reagents

A previously characterised rat CB1R construct in which the first 25 N-terminal amino acids were omitted to avoid possible cleavage of the SBP-EGFP tag (*McDonald et al., 2007b*; *Nordström and Andersson, 2006*) was used as a template for sub-cloning into pcDNA3.1. This construct displays normal plasma membrane trafficking (*McDonald et al., 2007b*; *Hebert-Chatelain et al., 2016*) and removes the region reported to constitute a mitochondrial targeting motif (*Hebert-Chatelain et al., 2016*). Helix 9 (residues 440–460) was removed by site-directed mutagenesis. These WT and ΔH9 constructs were subsequently used as a template to clone into the RUSH vector system (interleukin-2 signal peptide followed by SBP and EGFP N-terminal tags) as previously described (*Evans et al., 2017*; *Boncompain and Perez, 2013*). Non-ER-retained SBP-EGFP-tagged versions were obtained by re-cloning these inserts from the RUSH vector into pcDNA3.1 (pcDNA-SP$^{II-2}$-SBP-EGFP-CB1R). The SBP tag was deleted for surface biotinylation experiments by site-directed mutagenesis (pcDNA-SP$^{II-2}$-EGFP-CB1R). Chimeric CD4-ctCB1R WT and ΔH9 were generated by overlap extension PCR followed by cloning into a plasmid expressing CD4 lacking its own C-terminus (*Garrido, 2001*).

Chicken anti-GFP was from Abcam (ab13970); mouse anti-Ankyrin-G was from NeuroMab (clone N106/36); rabbit anti-MAP2 was from Synaptic Systems (188 003); mouse anti-CD4 was from BioLegend (clone OKT4); rat anti-GFP was from ChromoTek (3H9); anti-phosphoERK (M7802), and anti-non-phosphoERK (M3807) were from Sigma; mouse anti-GAPDH (6C5 ab8245) was from Abcam. All fluorescent secondaries were from Jackson Immunoresearch Laboratories and HRP conjugated secondaries were from Sigma. ACEA and AM281 were from Tocris bio-techne.

## Cell culture and transfection

Dissociated hippocampal cultures were prepared from E17-E18 Wistar rats as previously described (*Martin and Henley, 2004*). Glass coverslips were coated in poly-D-lysine or poly-L-lysine (1 mg/mL, Sigma) in borate buffer (10 mM borax, 50 mM boric acid) overnight and washed in water. Dissociated hippocampal cells were plated at different densities in plating medium (Neurobasal, Gibco supplemented with 10% horse serum, Sigma; 2 mM GlutaMAX, Gibco; and either GS21, GlobalStem, or B27, Thermo Fisher) which was changed to feeding medium (Neurobasal supplemented with 1.2 mM GlutaMAX and GS21 or B27) after 24 hr. For RUSH experiments, cells were plated and fed in media containing GS21 instead of B27 because it does not contain biotin. Cells were incubated at 37°C and 5% $CO_2$ for up to 2 weeks. Animal care and procedures were carried out in accordance with UK Home Office and University of Bristol guidelines.

Transfection of neuronal cultures was carried out at DIV 12 using Lipofectamine2000 (Invitrogen) according to the manufacturer's instructions with minor modifications. Cells were left for 20–48 hr before fixation.

HEK293T cells (ECACC) were passaged and maintained in complete DMEM (DMEM + 10% FBS + 2 mM L-Glutamine). HEK293T cells were regularly treated with ciprofloxacin (10 µg/mL) to prevent mycoplasma contamination.

## Phospho-ERK assay

HEK293T cells were transfected with SBP-EGFP-CB1R$^{WT}$, SBP-EGFP-CB1R$^{\Delta H9}$, or empty pcDNA3.1 and left for 24 hr. The cells were serum-starved overnight and then treated with 1 µM ACEA or 0.01% EtOH for 5 min before being lysed in lysis buffer (50 mM Tris-HCl; 150 mM NaCl; 1% CHAPS, ThermoFisher Scientific; protease inhibitors, Roche) with phosphatase inhibitors (Pierce, Thermo-Fisher Scientific). SDS-PAGE and Western blotting procedures were carried out according to standard protocols.

## Surface biotinylation assay

HEK293T cells were transfected with EGFP-CB1R$^{WT}$ or EGFP-CB1R$^{\Delta H9}$ for 48 hr, then cooled to 4°C on ice and washed 3 times in ice-cold PBS. 0.3 mg/mL of EZ-Link Sulfo-NHS-SS-Biotin in PBS was added for 10 min, then the cells were washed 3 times in PBS. 50 mM $NH_4Cl$ in PBS was added for 2 min to quench any remaining unreacted biotin, and the cells were washed another 3 times in PBS before being lysed in lysis buffer. Biotinylated surface proteins were isolated using streptavidin coated agarose beads according to standard protocols.

## Live surface staining and antibody feeding

To measure surface staining, cultured neurons were cooled at room temperature for 5–10 min, then incubated with the appropriate antibody (chicken anti-GFP or mouse anti-CD4) in conditioned media for 10–20 min at RT. The neurons were washed multiple times in PBS before fixation.

For agonist and inverse agonist experiments, the neurons were treated with 5 µM ACEA (in EtOH) or vehicle control (0.1% EtOH) for 3 hr or 10 µM AM281 (in DMSO) or vehicle control (0.2% DMSO) for 3 hr in conditioned media at 37°C and 5% $CO_2$, and then subsequently surface stained.

To measure endocytosed receptors, neurons were fed with the appropriate antibody (chicken anti-GFP or mouse anti-CD4) for 2 hr in conditioned media at 37°C and 5% $CO_2$. Neurons were washed several times in PBS and then surface antibody was stripped by two quick washes with ice-cold pH 2.5 PBS (anti-GFP) or a 4 min incubation with 0.5M NaCl and 0.2M acetic acid (anti-CD4) followed by several washes in PBS before fixation.

## RUSH live labelling

Neurons were transfected with RUSH constructs at DIV 12 for no longer than 24 hr to prevent ER stress resulting from accumulation of unreleased receptors. Neurons were incubated in conditioned media containing D-biotin (40 µM, Sigma) and chicken anti-GFP (1:1,000) for different lengths of time at 37°C and 5% $CO_2$. The 0 min timepoint was only incubated with chicken anti-GFP without biotin for 60 min. For the O/N timepoint, neurons were incubated in 40 µM D-biotin immediately following transfection and then left overnight at 37°C and 5% $CO_2$ before being incubated with biotin and chicken anti-GFP for 60 min to label surface CB1R. Every independent experiment included a 60

min timepoint to which values were normalised and a 0 min control. Following biotin treatment, neurons were washed several times in PBS and cooled to 4°C to prevent further internalisation. They were then live labelled with 647-labelled anti-chicken in conditioned media for 15 min at 4°C before being fixed and permeabilised and stained with Cy3-labelled anti-chicken. In the text, 'surface' thus refers to 647 fluorescence acquisition, whereas 'surface + endocytosed' refers to Cy3 fluorescence acquisition.

## Fixation and fixed immunostaining

Cultured neurons were fixed in 4% formaldehyde in PBS for 12 min, then washed 3x in PBS, 1x in 100 mM Glycine in PBS, and 3x in PBS. The neurons were then blocked and permeabilised in PBS + 3% BSA + 0.1% Triton X-100 before being incubated in fluorescent secondary (1:400) in PBS + 3% BSA. Subsequently, the neurons were re-incubated in primary antibody (anti-GFP or anti-CD4) to measure total levels of expression and stained with either anti-MAP2 (dendritic marker) or anti-Ankyrin-G (axonal initial segment marker) in PBS + 3% BSA. The neurons were then washed several times in PBS and mounted onto glass slides using Fluoromount-G (ThermoFisher Scientific).

## Image acquisition and analysis

Images were acquired using either a Leica SPE single channel confocal laser scanning microscope or a Leica SP8 AOBS confocal laser scanning microscope (Wolfson Bioimaging Facility, University of Bristol). All settings were kept the same within experiments. Neurons used for data acquisition were selected only on their total staining.

All quantification was performed using FIJI (ImageJ) software. Based on previous experiments, at least five cells were analysed per experiment, and at least three independent experiments (i.e. on different neuronal cultures on different days) were performed.

Images were max projected, and regions of interest (ROIs) of approximately similar lengths were drawn around axons and 3–4 proximal and secondary dendrites based on the total channel only. Axons were defined either as processes whose initial segment was positive for Ankyrin-G or as processes negative for MAP2. The mean fluorescence was measured for each channel and the dendritic values were averaged. 'Surface' or 'endocytosed' mean fluorescence values were normalised to the 'total' mean fluorescence value for each ROI to account for varying levels of expression of transfected constructs. These values were then normalised to the axon value of the control (WT or CD4). For drug treatments, surface values were normalised to their respective vehicle treated controls and sampled at the same time-point (i.e. WT + drug was normalised to WT + vehicle and ΔH9 + drug was normalised to ΔH9 + vehicle) to account for possible differences in steady-state surface expression between the constructs and/or constitutive internalisation.

Because of the change in total mean fluorescence in axons throughout the different conditions, the above image analysis was slightly modified for RUSH experiments. In these experiments, neurites were traced using NeuronJ so that only the mean fluorescence of exactly the first 50 μm of the axons and 30–40 μm of 2–4 primary dendrites for each channel was measured. All 'surface' and 'surface + endocytosed' values (of both axons and dendrites) were normalised to the average total dendritic value for each neuron. Axon total mean fluorescence was also normalised to the average total dendritic value within each cell. All values were then normalised to the WT 60 min axon value within each experiment. In a slightly smaller subset of RUSHed neurons, axons were traced for ~100 μm using NeuronJ. Line plots were generated and the mean fluorescence was averaged in 10 μm segments. The averages for each 10 μm segment from each cell were normalised first to the dendritic total value, then to the axonal 60 min value of the first 50 μm.

Polarity indices (A/D ratio) were calculated by dividing the axonal mean fluorescence value by the average dendritic mean fluorescence value.

The scalebar for all images represents 20 μm.

## Statistics

All statistics were performed using GraphPad Prism. The ROUT method was used to identify outliers for all parameters measured before normalising to control. Neurons were removed from analysis if any one parameter was found to be an outlier. As is the convention in the field (*Leterrier et al., 2006*; *Coutts et al., 2001*; *Simon et al., 2013*; *Evans et al., 2017*; *McDonald et al., 2007b*;

*Leterrier et al., 2017*), 'N' denotes the number of separate neuronal cultures prepared from litters of pups from separate dams and 'n' denotes the total number of neurons across the separate cultures assessed. To determine statistical significance between two groups, a D'Agostino and Pearson normality test was performed. Unpaired t-tests were performed on data that passed the normality test whereas the Mann-Whitney test was used if it did not. One- or Two-way ANOVAs with Tukey's or Sidak's *post hoc* test were used to determine statistical significance between more than two groups depending on the comparisons required. *$p \leq 0.05$, **$p \leq 0.01$, ***$p \leq 0.001$, ****$p \leq 0.0001$. All data are presented as mean $\pm$ SEM.

## Acknowledgements

We are grateful to the Wellcome Trust for financial support. AFJ is funded by a University of Bristol PhD Scholarship. We thank F Perez and G Boncompain (Institut Curie, Paris) for the RUSH constructs, A Irvine (University of Dundee) for template CB1R plasmids and B Dargent (Université de la Mediterranée, Marseille) for CD4 plasmids. We gratefully acknowledge the excellent Wolfson Bioimaging Facility at the University of Bristol, supported by the BBSRC.

## Additional information

### Funding

| Funder | Author |
| --- | --- |
| Wellcome | Ashley J Evans |
| Medical Research Council | Kevin A Wilkinson<br>Jeremy M Henley |
| Biotechnology and Biological Sciences Research Council | Kevin A Wilkinson<br>Jeremy M Henley |
| University of Bristol | Alexandra Fletcher-Jones |

The funders had no role in study design, data collection and interpretation, or the decision to submit the work for publication.

### Author contributions

Alexandra Fletcher-Jones, Conceptualization, Data curation, Formal analysis, Investigation, Methodology, Writing—review and editing; Keri L Hildick, Conceptualization, Investigation, Methodology; Ashley J Evans, Conceptualization, Methodology, Project administration, Writing—review and editing; Yasuko Nakamura, Resources, Methodology, Project administration; Kevin A Wilkinson, Conceptualization, Supervision, Funding acquisition, Writing—review and editing; Jeremy M Henley, Conceptualization, Supervision, Funding acquisition, Writing—original draft, Project administration, Writing—review and editing

### Author ORCIDs

Alexandra Fletcher-Jones http://orcid.org/0000-0002-6192-2901
Ashley J Evans http://orcid.org/0000-0002-6658-2176
Kevin A Wilkinson https://orcid.org/0000-0002-8115-8592
Jeremy M Henley http://orcid.org/0000-0003-3589-8335

### Ethics

Animal experimentation: The use of animals is unavoidable because there is no satisfactory alternative preparation for studying neuronal and synaptic mechanisms that do not involve the use of tissue acutely removed from animals. All experiments in this study were designed to ensure the use of minimum numbers of animals which were sacrificed by cervical dislocation in strict accordance with UK Home Office Schedule 1 guidelines using procedures approved by the Animal Welfare and Ethics Review Body (AWERB) at the University of Bristol (approval reference number UIN UB/18/004).

Decision letter and Author response
Decision letter https://doi.org/10.7554/eLife.44252.014
Author response https://doi.org/10.7554/eLife.44252.015

## Additional files

### Supplementary files
• Transparent reporting form
DOI: https://doi.org/10.7554/eLife.44252.012

### Data availability
All relevant data are included in this published article. None of the data was suitable for depositing in repositories and no new software was generated.

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
