## [Decision Letter]

Thank you for submitting your article "The C-terminal Helix 9 motif regulates cannabinoid receptor type 1 trafficking and surface expression" for consideration by *eLife*. Your article has been reviewed by Richard Aldrich as the Senior Editor, a Reviewing Editor, and three reviewers. The following individuals involved in review of your submission have agreed to reveal their identity: Zsolt Lenkei (Reviewer #2).

The reviewers have discussed the reviews with one another and the Reviewing Editor has drafted this decision to help you prepare a revised submission

Summary:

The manuscript of Fletcher-Jones et al. investigates the molecular mechanisms of a prototypical axonal GPCR, the CB1 cannabinoid receptor. In addition to re-investigating several debated previous findings of CB1R targeting, such as the role of receptor-state dependent somatodendritic endocytosis, their study introduces a novel element, the role of the C-terminal segment in general and its Helix 9 (H9) in particular, by suggesting the involvement of H9 in specific targeting to axons. The use of the RUSH protocol to examine early steps of CB1R targeting is interesting. However, several shortcomings limit the enthusiasm for the manuscript.

Essential revisions:

1) The CB1 receptor displays a well-described disto-proximal gradient of expression at the axonal plasma membrane both in vitro and in vivo, with the highest receptor levels present at the pre-terminal or distal axonal segments. Given the highly branched morphology of typical CB1 expressing neurons, correct axonal polarization likely requires specific distal targeting mechanisms and not simply diffusion on the membrane from the proximal axonal surface. Unfortunately, the present manuscript investigates CB1R targeting only to a short and highly proximal axonal region, the first 50µm that also contains the axon initial segment (AIS), a highly-specialized axonal sub-region. This proximal segment is not representative for entire axonal surface, which includes the functionally more relevant distal segments. So while it's possible that this proximal axonal segment specifically captures CB1Rs delivered through H9-dependent transport in the early stages of CB1R export, it seems premature to draw conclusions on the mechanisms of polarized axonal expression of CB1Rs, which was reported to be established at a longer time-scale of several hours (Figure 7 in Leterrier et al., 2006), through measurement of CB1 targeting only to this sub-region during the first 90 minutes of release from the ER.

2) The data presented on Figure 2 on the effect of the C-terminal segment of CB1R on CD4 expression are interesting. The authors suggest specific axonal transport of the chimeric proteins. However, the green color on the merged images of CD4-ctCB1R constructs suggests that they are mostly endocytosed, while the WT CD4 is mostly surface-localized, as expected for a surface-expressed protein. Given the known, and in this manuscript partially confirmed the role of endocytosis in CB1R targeting to the axon, an alternative hypothesis is that the C-terminal segment of CB1R is driving CD4 into more endocytosed phenotype at least in dendrites, contributing to its axonal polarization. Consequently, the role of ctCB1R^WT^ and ctCB1R^∆H9^ in CD4 endocytosis should be excluded/analyzed more precisely.

3) On Figure 5A-C, reduced activation of ERK 1/2 is demonstrated for EGFP-CB1R^∆H9^ as compared to EGFP-CB1R^WT^ after normalizing for the amount of expressed receptors. However, since the *∆H9* mutants were shown to be more endocytosed in neurons, it is important to normalize rather for the surface population of receptors, since endocytosed EGFP-CB1R^∆H9^ may be less accessible to exogenous ligands, putatively explaining the measured differences in signaling.

4) The authors use an N-terminally truncated CB1R as "CB1R^WT^". This choice should be explained in the manuscript and the lack of effect of this truncation on axonal polarization should be shown, either by citing the literature or by experimental demonstration.

5) I have main concerns around the use of the statistics, I have not come across N vs. n used in these standardized test before. It appears that the authors have combined data from each individual neuron in order to carry out the statistics, as opposed to using an n=3 from the independent experiments, or some form of nested analysis where each experiment is treated separately. Perhaps this is the convention for this field, but it needs to be more clearly explained and justified.

6) HEK cells were used to look at signaling – why not do the pERK assay in neurons? While the cells appear to express similar total receptor was there equivalent cell surface expression? (Figure 5C).

7) The internalization assay was carried out at 3 hours which is enough time for significant constitutive internalization to have occurred. Were these compared to vehicle treatment or to time zero? A time course of internalization (which could have been done on the HEK cells) would provide greater insight into the mechanism by which this relatively subtle difference is occurring (the data sets are very much overlapping – see concerns around n vs. N in point one).

8) The use of the RUSH system is very powerful in helping to interpret the movement of the receptors – have the authors established that the tags on the receptor do not significantly alter the expression and function of the receptor (once released from the ER)?

9) In the brain little CB1 is seen along axons – rather it is very localized to the presynaptic terminal, yet in cells in culture the entire axon appears to express surface CB1 – does this suggest that trafficking is different in culture to in vivo?

---

## [Author Response]

Essential revisions:1) The CB1 receptor displays a well-described disto-proximal gradient of expression at the axonal plasma membrane both in vitro and in vivo, with the highest receptor levels present at the pre-terminal or distal axonal segments. Given the highly branched morphology of typical CB1 expressing neurons, correct axonal polarization likely requires specific distal targeting mechanisms and not simply diffusion on the membrane from the proximal axonal surface. Unfortunately, the present manuscript investigates CB1R targeting only to a short and highly proximal axonal region, the first 50µm that also contains the axon initial segment (AIS), a highly-specialized axonal sub-region. This proximal segment is not representative for entire axonal surface, which includes the functionally more relevant distal segments. So while it's possible that this proximal axonal segment specifically captures CB1Rs delivered through H9-dependent transport in the early stages of CB1R export, it seems premature to draw conclusions on the mechanisms of polarized axonal expression of CB1Rs, which was reported to be established at a longer time-scale of several hours (Figure 7 in Leterrier et al., 2006), through measurement of CB1 targeting only to this sub-region during the first 90 minutes of release from the ER.

We thank the reviewers for raising this key issue and we apologise that we were not sufficiently clear in our original submission. We have added additional data and text to clarify the revised manuscript.

We used time-resolved RUSH experiments to investigate the initiation of CB1R polarity. We measured the transit through the secretory pathway and incorporation into, and passage through, the highly organised axon initial segment (AIS) that acts as a ‘gate-keeper’ for proteins entering the axonal compartment. Our data show that CB1R polarisation is initiated in the first 90 min since they are directly targeted to, and surface expressed within, proximal axonal regions.

We have edited our revised manuscript to explicitly state that our time-resolved experiments were designed to examine the initiation of polarity and that we chose to examine the most proximal region of the axon to establish when CB1Rs first reach the axon.

Importantly, we also now emphasize that the mechanism for the initiation of polarity that we describe here does not exclude the possibility that further trafficking signals determine the subsequent distal targeting and presynaptic enrichment of the receptor. These distal CB1R trafficking processes may include lateral diffusion and/or trafficking of CB1R-containing vesicles originating from the secretory pathway or transcytosis via the endosomal system (Simon et al., 2013) inside the axon. As the reviewer suggests, these additional mechanisms may take place on a time-scale of several hours.

To further address this issue, we performed additional analyses to monitor CB1R delivery beyond the AIS. We traced the first 100 µm of axon from the soma and plotted total and surface CB1R delivery at 0, 30, 90 minutes and overnight (O/N) after biotin-mediated release from the ER (Figure 2H,I). Interestingly, 30 minutes after release from the ER, before CB1R appears on the surface, CB1R is already present at least 100 µm along the axon at levels similar to the effectively unretained control (O/N). Furthermore, surface CB1Rs delivered from the secretory pathway accumulate at the distal region of the AIS before progressing further along the axon. 100 µm along the axon, CB1R levels reach a steady state 90 min after release, again at levels similar to the O/N control. However, when released O/N, fewer receptors remain in the most proximal region of the axon, consistent with the involvement of other mechanisms in distal axonal delivery (>100 µm) over a time-course of several hours.

Overall, we interpret these data to suggest that arrival at the AIS likely represents an initial ‘axonal capture’ of CB1R. Once within the axonal compartment trafficking to more distal locations and to presynaptic sites is then mediated by additional mechanisms that probably include both intracellular transport and lateral diffusion and trapping within the axonal membrane analogous to processes reported for postsynaptic AMPA receptors (Penn et al., 2017). While of undoubted interest, we believe that the detailed investigation of these subsequent distal axonal trafficking steps for CB1R is beyond the scope of our current manuscript.

2) The data presented on Figure 2 on the effect of the C-terminal segment of CB1R on CD4 expression are interesting. The authors suggest specific axonal transport of the chimeric proteins. However, the green color on the merged images of CD4-ctCB1R constructs suggests that they are mostly endocytosed, while the CD4^WT^ is mostly surface-localized, as expected for a surface-expressed protein. Given the known, and in this manuscript partially confirmed the role of endocytosis in CB1R targeting to the axon, an alternative hypothesis is that the C-terminal segment of CB1R is driving CD4 into more endocytosed phenotype at least in dendrites, contributing to its axonal polarization. Consequently, the role of ctCB1R^WT^ and ctCB1R^∆H9^ in CD4 endocytosis should be excluded/analyzed more precisely.

In the original manuscript we analysed differences in endocytosis between full-length SBP-EGFP-CB1R^WT^ and SBP-EGFP-CB1R^ΔH9^. Crucially, the only the difference between these constructs is omission of the H9 motif within the C-terminal domain of the receptor. These experiments demonstrate that the H9 domain plays a role in stabilising surface expression at the axonal membrane and that SBP-EGFP-CB1R^ΔH9^ is more rapidly endocytosed than SBP-EGFP-CB1R^WT^ in both axons and in dendrites. Moreover, we showed that the CB1R-specific inverse agonist AM281 significantly reduced surface polarity of EGFP-CB1R^ΔH9^ compared to EGFP-CB1R^WT^ in hippocampal neurons due to increased dendritic surface expression (Figure 8).

As suggested, to more fully address the role of H9 in endocytosis we have now performed additional antibody feeding experiments using the CD4-ctCB1R^WT^ and ctCB1R^ΔH9^ fusion proteins in neurons and added these new data to the revised manuscript (Figure 4). Entirely consistent with the AM281 results from full-length CB1Rs there is no difference in internalisation between CD4-ctCB1R^WT^ and ctCB1R^ΔH9^ in either axons or dendrites, since the CD4 chimeras are incapable of endocannabinoid-induced internalisation.

Interestingly, there is a slight, but significant, increase in dendritic internalisation of both CD4-ctCB1R^WT^ and ctCB1R^ΔH9^ compared to the CD4^Δct^ control. Importantly, because this increase in internalisation in identical between CD4-ctCB1R^WT^ and ctCB1R^ΔH9^, it cannot account for the failure of CD4-ctCB1R^ΔH9^ to polarise to the level of CD4-ctCB1R^WT^. However, it could provide an explanation for why CD4-ctCB1R^ΔH9^ surface polarisation does not return completely to CD4^Δct^ control levels.

This dendrite selective increase in internalisation suggests the presence of a constitutive activation-independent internalisation mechanism that does not involve H9 and is absent from the axonal compartment. Although detailed identification of this process is beyond the scope of the current manuscript, we have included new discussion to highlight this point.

3) On Figure 5A-C, reduced activation of ERK 1/2 is demonstrated for EGFP-CB1R^∆H9^ as compared to EGFP-CB1R^WT^ after normalizing for the amount of expressed receptors. However, since the ∆H9 mutants were shown to be more endocytosed in neurons, it is important to normalize rather for the surface population of receptors, since endocytosed EGFP-CB1R^∆H9^ may be less accessible to exogenous ligands, putatively explaining the measured differences in signaling.

We have now used surface biotinylation to assess surface expression of EGFP-CB1R^WT^ and EGFP-CB1R^ΔH9^ in HEK cells. We detected no difference, indicating that EGFP-CB1R^ΔH9^ is not more internalised in this cell line. Furthermore, these data demonstrate that the lack of ERK activation by EGFP-CB1R^ΔH9^ is attributable to a deficit in signalling capability rather than altered surface expression. We have added this new data to the revised manuscript (Figure 7D-E).

4) The authors use an N-terminally truncated CB1R as "CB1R^WT^". This choice should be explained in the manuscript and the lack of effect of this truncation on axonal polarization should be shown, either by citing the literature or by experimental demonstration.

We thank the reviewer for this point, and we apologise for not including these details in the original manuscript.

N-terminal fluorphore-tagged-CB1R was first reported by the Irving group (McDonald et al., 2007). In that construct the extreme 25 N-terminal amino acids were removed to avoid possible cleavage of the tag during protein processing (Nordstrom et al., 2006). Interestingly, it has subsequently been reported that the first 22 residues constitute a mitochondrial targeting motif, and that removal of these 22 residues prevents mitochondrial localisation of CB1R but does not affect plasma membrane trafficking (Hebert-Chatelain et al., 2016).

In all reports, the plasma membrane trafficking and axonal polarisation of N-terminally truncated CB1 was indistinguishable from endogenous CB1R. Thus, our use of the N-terminal truncations circumvents potential complications due to mitochondrial targeting which, although of interest, are not the focus of our study. Nonetheless, to fully confirm these reports we generated an entirely full-length EGFP-CB1R (EGFP-CB1R^FL^) that is not missing the N-terminal amino acids. As shown in Author response image 1, the axonal surface polarisation of this full-length CB1R behaves identically to the truncated version used throughout our manuscript. These points are now included in the revised manuscript.

**Author response image 1. respfig1:** Indistinguishable axonal surface polarisation of full-length CB1R and CB1R lacking the first 25 N-terminal residues.

5) I have main concerns around the use of the statistics, I have not come across N vs. n used in these standardized test before. It appears that the authors have combined data from each individual neuron in order to carry out the statistics, as opposed to using an n=3 from the independent experiments, or some form of nested analysis where each experiment is treated separately. Perhaps this is the convention for this field, but it needs to be more clearly explained and justified.

As now indicated in the text, the analyses we have used are indeed the convention for the field and have been used in multiple papers including (Soltesz et al., 2015; Basavarajappa et al., 2017; Robin et al., 2018; Irving et al., 2000).

6) HEK cells were used to look at signaling – why not do the pERK assay in neurons? While the cells appear to express similar total receptor was there equivalent cell surface expression? (Figure 5C).

Performing the pERK assay in neurons is complex because of the confounding issue of endogenous CB1R expression. To be meaningful, experiments would require knockdown of endogenous CB1R and subsequent replacement with EGFP-CB1R^WT^ or EGFP-CB1R^ΔH9^. While this would be feasible in the long term, we argue it is unnecessary for these fundamental signalling experiments. Rather, because HEK cells don’t contain endogenous CB1R but do contain all the components of the ERK pathway, they provide a convenient, tractable, and standardly used model system to test the ability of EGFP-CB1R^WT^ and EGFP-CB1R^ΔH9^ to couple to pERK signalling.

As set out in our response to point 3, we have performed surface biotinylations to confirm that EGFP-CB1R^WT^ and EGFP-CB1R^ΔH9^ are surface expressed at the same levels in HEK293T cells. These data are now included in the revised manuscript (Figure 7D-E).

7) The internalization assay was carried out at 3 hours which is enough time for significant constitutive internalization to have occurred. Were these compared to vehicle treatment or to time zero? A time course of internalization (which could have been done on the HEK cells) would provide greater insight into the mechanism by which this relatively subtle difference is occurring (the data sets are very much overlapping – see concerns around n vs. N in point one).

As the reviewer suggests, results were compared to a vehicle treatment control sampled at the same time-point, which controls for any constitutive internalisation. More specifically, EGFP-CB1R^WT^ and + ACEA was normalized to EGFP-CB1R^WT^ + EtOH, and EGFP-CB1R^ΔH9^ + ACEA was normalized to EGFP-CB1R^ΔH9^ + EtOH to fully account for any differences in steady-state levels of surface expression between the constructs and for any differences in constitutive internalisation. This is now better explained in the revised manuscript.

8) The use of the RUSH system is very powerful in helping to interpret the movement of the receptors – have the authors established that the tags on the receptor do not significantly alter the expression and function of the receptor (once released from the ER)?

As outlined above, N-terminal tags do not impede CB1R plasma membrane localisation or function. Moreover, our pERK signalling assays further validate and confirm the functionality of the SBP-EGFP-CB1R constructs (Figure 7).

9) In the brain little CB1 is seen along axons – rather it is very localized to the presynaptic terminal, yet in cells in culture the entire axon appears to express surface CB1 – does this suggest that trafficking is different in culture to in vivo?

This is a difficult question to address directly but we very much doubt that different trafficking mechanisms are employed in culture to those that occur in the brain. We note, however, that the pattern of localisation we observe is entirely consistent with previous studies investigating the distribution of endogenous and expressed CB1R in cultured neurons (Soltesz et al., 2015; Basavarajappa et al., 2017; Busquets-Garcia et al., 2018: Coutts et al., 2001).

We agree that CB1R localisation in brain sections, determined mainly using immunogold electron microscopy, predominantly detects CB1R at the presynaptic terminal (Katona, 2009; Simon et al., 2013). However, immunogold staining detects only a relatively small fraction of the total antigen present (Rozenfeld and Devi, 2008; Rozenfeld, 2011). Therefore, because CB1Rs are concentrated in the bouton, that is the region where they are most reliably visualised. A recent paper using STORM super-resolution imaging of CB1R in brain slices specifically focused on presynaptic boutons but did not compare this region to the axonal shaft (Hsieh et al., 1999). Furthermore, the fact that dispersed neuronal cultures are more tractable to manipulation and permeabilization, and therefore likely more accessible to antibodies than brain slices, may also partly explain why axonal CB1Rs are more readily observed in culture.

We emphasize, however, that CB1Rs present at presynaptic termini must initially be delivered and therefore must undergo targeting to, and trafficking via, the axon. Importantly, it is aspects of these processes that we are investigating.

References:

Penn AC, Zhang CL, Georges F, Royer L, Breillat C, Hosy E, et al. Hippocampal LTP and contextual learning require surface diffusion of AMPA receptors. Nature. 2017;549(7672):384-8.